# UnitNorm: Rethinking Normalization for Transformers in Time Series

## Abstract

Normalization techniques are crucial for enhancing Transformer models' performance and stability in time series analysis tasks, yet we originally identify that traditional methods like batch and layer normalization often lead to issues such as token shift, attention shift, and sparse attention. We propose UnitNorm, a novel normalization approach that scales input vectors by their norms and modulates attention patterns, effectively circumventing these challenges. Grounded in existing normalization frameworks, UnitNorm's effectiveness is demonstrated across diverse time series analysis tasks, including forecasting, classification, and anomaly detection, via a rigorous evaluation on 6 state-of-the-art models and 10 datasets. UnitNorm demonstrates superior performance, particularly where robust attention and contextual understanding are vital, achieving up to a 1.46 MSE decrease in forecasting and a 4.89% accuracy increase in classification. This work not only calls for a re-evaluation of normalization strategies in time series Transformers but also sets a new direction for enhancing model performance and stability. The source code is available at https://anonymous.4open.science/r/UnitNorm-5B84.

## 1 Introduction



Figure 1: Scheme of different normalization methods applied to batched sequences of **time series tokens** $\mathbf{X} \in \mathbb{R}^{N \times L \times D}$, where $N$ is the batch size, $L$ is the sequence length (or historical window length) and $D$ is the feature dimension (number of variates) of each token vector. The blue sections demonstrate a single slice of the input tensor for computing the mean $\mu$ and variance $\sigma^2$, while the red section shows a single slice of data for computing the vector norm $\|\mathbf{x}\|$ (see Section C.1).

Transformers have emerged as powerful tools for time series analysis (TSA), offering new capabilities for modeling complex temporal dependencies (Wen et al., 2023). However, adapting them from domains like NLP (Wolf et al., 2020) and CV (Han et al., 2023) presents challenges, especially regarding how normalization interacts with time series characteristics like **seasonality, trends, and autocorrelations**. Central to these models is the representation of data as sequences of tokens, denoted by $\mathbf{X} \in \mathbb{R}^{N \times L \times D}$, where $N$ stands for batch size, $L$ is the sequence length and $D$ represents the dimensionality of each token.

Time series data presents unique challenges that distinguish it from other domains. These include capturing **complex/multi-scale temporal dependencies**, **periodic patterns**,

and **variable sampling frequencies**. Unlike text or images, time series often exhibit **strong autocorrelations**, **seasonal/trend components**, and **non-stationarity**. These properties mean data distributions can shift significantly, making normalization easily distortive. While Transformers have shown promise in time series forecasting, classification, and anomaly detection, they were not originally designed with these specific characteristics of time series data in mind. This is particularly evident in how normalization techniques, developed for more stationary data, interact with and potentially disrupt the learning of temporal patterns and periodic structures.

The core mechanism facilitating the Transformers' ability to model complex dependencies is the attention mechanism. It computes a weighted sum of value vectors $\mathbf{V}$, capturing the sequential relationships between tokens through a scalable dot-product operation of queries $\mathbf{Q}$ and keys $\mathbf{K}$: $\text{Attention}(\mathbf{Q}, \mathbf{K}, \mathbf{V}) = \text{softmax}\left(\frac{\mathbf{Q}\mathbf{K}^{\top}}{\sqrt{D}}\right)\mathbf{V}$. Vaswani et al. (2017)

To mitigate issues during the training process of Transformers related to vanishing or exploding gradients Lubana et al. (2021); Yang & Schoenholz (2017), Layer Normalization (LayerNorm, LN, Ba et al. 2016) plays a significant role and is therefore incorporated at each sub-layer of the architecture (Figure S1)[1]. The LayerNorm operation follows the center-and-scale standardization paradigm, by first centering the means to 0 and then rescaling the variances of the input vectors to 1 such that $\text{LN}(\mathbf{X}) = \frac{\mathbf{X} - \boldsymbol{\mu}}{\sqrt{\boldsymbol{\sigma}^2 + \varepsilon}}$, where $\boldsymbol{\mu}$ and $\boldsymbol{\sigma}$ are the mean and standard deviation of the input vector $\mathbf{X}$, respectively. Ba et al. (2016)

While LayerNorm, compared to other normalization strategies such as batch normalization Ioffe & Szegedy (2015); Shen et al. (2020); Wang et al. (2022), has established itself as the dominant normalization strategy in Transformers, dedicated normalization-specific research has mostly focussed on its impact on model convergence Wang et al. (2019), its inner dynamics Wang et al. (2022); Shen et al. (2020) or its location Xiong et al. (2020) within the architecture. On the other hand, few works address normalization's interaction with the attention mechanism Kobayashi et al. (2021) (Section 5), a key challenge in TSA (Section 2) due to attention's dot product.

In this work, we provide a new viewpoint on these challenges by first identifying and formalizing Transformer-specific challenges of normalization techniques, highlighting three key issues. Building on these insights, we introduce a novel normalization technique, UnitNorm, designed to address these challenges effectively.

Our contributions lie in: 1) We **originally identify** two challenges, namely *token shift* and *attention shift*, and reassess the challenge of *sparse attention* Zhai et al. (2023) in Transformers for time series analysis; 2) We propose a new normalization method, UnitNorm, that can mitigate these issues by design, thereby **better preserving crucial temporal information**; 3) We empirically validate the effectiveness of UnitNorm on nine datasets spanning three downstream TSA tasks.

## 2   Challenges in Normalization

Transformers rely on attention mechanisms to achieve remarkable performance in time series analysis tasks. However, the interplay between attention and normalization methods introduces critical, unaddressed challenges. This paper aims to reveal the complexities of token shift, attention shift, and sparse attention, which arise from such interaction between normalization and the attention mechanism. Our theoretical and empirical analysis demonstrates these challenges are intrinsic to conventional normalization, impacting self-attention in time series Transformers.

Time series data presents distinct challenges for Transformers due to its inherent temporal properties. Unlike text or image data, time series often contain critical periodic patterns, trends, and seasonal components that require a balanced attention distribution to capture

---

[1]The LayerNorm used in Transformers, referred to as LayerNorm (practice), computes the statistics within each token rather than over the whole batch as LayerNorm (theory) does (Figure 1). In this paper, we will refer to the LayerNorm (practice) as LayerNorm if no distinction is made.

Table 1: Effect of input transformations on the softmax function output. Importance order invariant refers to whether the relative importance of the tokens is preserved. Of all possible input transformations, only the reflection transformation will definitely change the importance order of the tokens.

| Type | Function | Input | | | Output | | | Order invariant? |
|------|----------|-------|---|---|--------|---|---|------------------|
| None | $f : x \mapsto x$ | -2 | 1 | 3 | 0.01 | 0.12 | 0.88 | |
| Stretch | $f : x \mapsto k \cdot x, k \in \mathbb{R}^+$ | -4 | 2 | 6 | 0 | 0.02 | 0.98 | ✓ |
| Translate | $f : x \mapsto x + a, a \in \mathbb{R}$ | -1 | 2 | 4 | 0.01 | 0.12 | 0.88 | ✓ |
| Jitter | $f : x \mapsto x + \varepsilon, \varepsilon \sim \mathcal{N}\left(0, \sigma^2\right)$ | -2.1 | 1.1 | 3 | 0.01 | 0.13 | 0.87 | ✓/✗ |
| Reflection | $f : x \mapsto -x$ | 2 | -1 | -3 | 0.95 | 0.05 | 0.01 | ✗ |

effectively. Conventional normalization, effective elsewhere, can disrupt these temporal relationships, *e.g.*, by altering token vector orientations and obscuring long-range dependencies. This disruption is particularly problematic in applications like forecasting periodic signals, detecting anomalies in regular patterns, or classifying time series based on their temporal characteristics—all tasks relying on accurately interpreting temporal token relationships.

## 2.1 Pilot Study: Normalization Impact on Capturing Periodicity

To illustrate normalization's impact on capturing periodicity—a key aspect of TSA—we conducted a pilot study on a synthetic two-channel sine wave dataset with varying periods, amplitudes, and Gaussian noise (details and full results in Table S2).

In this study (Table S2), UnitNorm ($k = 0.5$) drastically reduced MSE by 58.6% (from 2.721 for BatchNorm to 1.127) compared to other methods. This strongly illustrates how **traditional techniques can distort attention, via mechanisms like token shift and by inducing sparse attention** (detailed in Sections 2.2 to 2.4), thus impairing the capture of vital periodic patterns. Whereas UnitNorm's design, by **preserving token importance and promoting balanced attention** (Section 3), directly addresses these distortions.

The subsequent sections will now dissect these challenges, token shift (Section 2.2), attention shift (Section 2.3), and sparse attention (Section 2.4), both theoretically and empirically, before detailing UnitNorm's methodology (Section 3).

We will further explore the relationship between normalization and attention by examining a simplified equivalent attention process, with normalization preceding attention (Zhang et al. 2022, Figure S1). This perspective allows for a detailed exploration of how normalization influences the attention scores derived from the query and key vectors. For simplicity, our discussion will center on a singular instance of self-attention within the encoder layer, assuming identical query and key vectors to streamline our analysis (see Section C.2).

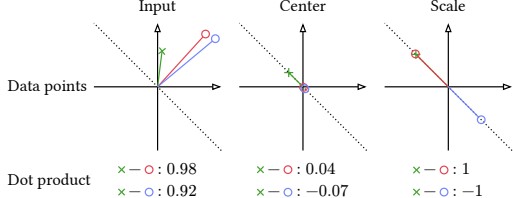

Figure 2: Case of token shift in LayerNorm. The green cross denotes a query vector, the red and blue circles denote two key vectors. Token shift at the centering step causes dot product sign flips; scaling does not.

## 2.2 Token shift

Previous study Brody et al. (2023) has attributed LayerNorm's efficacy to its center-and-scale operations: centering projects the input vectors to a hyperplane orthogonal to $\mathbb{1}$ vector, and scaling normalizes the vectors to a unit sphere to prevent any token vector being contained in the convex hull of the others. However, this can significantly alter the orientation of input vectors, especially for those that are near parallel to the hyperplane's norm vector $\mathbb{1}$. This impacts dot products, potentially causing sign flips (Figure 2), severely disrupting softmax

Table 2: Effect of normalization on the attention weight distribution based on empirical results (Figures S6 and S7). UnitNorm shows the most faithful representation of the original attention weights that are cross-validated by various metrics as described in Table S10, while center-and-scale normalization significantly alters the attention weights to an extreme extent as depicted in Figure S5.

| Normalization | Chebyshev distance ↓ | Cosine similarity ↑ | KL divergence ↓ | Entropy ↑ |
|---|---|---|---|---|
| None (original) | / | / | / | High |
| Center-and-scale | High | Low | High | Very Low |
| UnitNorm | Low | High | Low | High |

outputs (Table 1), and **catastrophically altering token importance** (Table 2). This issue of significant deviations in attention weight distributions caused by token shift will be further explored in Section 2.3.

The high probability of "center-and-scale" normalization inducing such sign flips is not merely theoretical, as Theorem 2.1 elucidates. Proof in Section B.

**Theorem 2.1** (High probability of sign flip due to center operation). *Assume that* $\mathbf{x} \sim \mathcal{N}(\boldsymbol{\mu}_x, \mathrm{diag}(\boldsymbol{\sigma}_x^2))$, $\mathbf{y} \sim \mathcal{N}(\boldsymbol{\mu}_y, \mathrm{diag}(\boldsymbol{\sigma}_y^2))$ *are two independent token vectors, with* $\boldsymbol{\mu}_x, \boldsymbol{\mu}_y, \boldsymbol{\sigma}_x, \boldsymbol{\sigma}_y \in \mathbb{R}^D$. *Let* $\tilde{\mathbf{x}} = \frac{\mathbf{x}-\boldsymbol{\mu}_x}{\boldsymbol{\sigma}_x}$ *and* $\tilde{\mathbf{y}} = \frac{\mathbf{y}-\boldsymbol{\mu}_y}{\boldsymbol{\sigma}_y}$ *be the normalized vectors. If*

$$
|\boldsymbol{\mu}_x^\top \boldsymbol{\mu}_y| \geq 12 \left( \sqrt{\boldsymbol{\sigma}_x^{2\top}\boldsymbol{\sigma}_y^2} + \|\boldsymbol{\sigma}_x \circ \boldsymbol{\sigma}_y\|_\infty \right) +
$$
$$
5 \left( \sqrt{\boldsymbol{\sigma}_y^{2\top}\boldsymbol{\mu}_x^2} + \sqrt{\boldsymbol{\sigma}_x^{2\top}\boldsymbol{\mu}_y^2} + \|\boldsymbol{\sigma}_y \circ |\boldsymbol{\mu}_x|\|_\infty + \|\boldsymbol{\sigma}_x \circ |\boldsymbol{\mu}_y|\|_\infty \right) \tag{1}
$$

*then the probability that the signs of* $\mathbf{x}^\top\mathbf{y}$ *and* $\tilde{\mathbf{x}}^\top\tilde{\mathbf{y}}$ *do not coincide is at least 40%, i.e.,*

$$
\Pr(\mathrm{sgn}(\mathbf{x}^\top\mathbf{y}) \neq \mathrm{sgn}(\tilde{\mathbf{x}}^\top\tilde{\mathbf{y}})) \geq 0.40. \tag{2}
$$

*Remark* 2.2. Derived from the computational methodologies for the statistics of vectors $\mathbf{x}$ and $\mathbf{y}$ (Section C.1), BatchNorm posits that the mean vectors are the same so that $\boldsymbol{\mu}_x = \boldsymbol{\mu}_y = \boldsymbol{\mu}$, and similarly $\boldsymbol{\sigma}_x^2 = \boldsymbol{\sigma}_y^2 = \boldsymbol{\sigma}^2$, while LayerNorm assumes that the mean and standard deviation are shared across feature dimension: $\boldsymbol{\mu}_x = \mu_x\mathbb{1}, \boldsymbol{\mu}_y = \mu_y\mathbb{1}$ and $\boldsymbol{\sigma}_x^2 = \sigma_x^2\mathbb{1}, \boldsymbol{\sigma}_y^2 = \sigma_y^2\mathbb{1}$. Given these assumptions, the condition (1) outlined in Theorem 2.1 is satisfied for many token vector distributions. In fact, we show that in the setup of LayerNorm, the condition (1) allows for the quotients of token means and standard deviations, i.e., for $\mu_x/\sigma_x$ and $\mu_y/\sigma_y$, to decay as $\Omega(D^{-1/4})$ while still implying a high sign flip probability, cf. Section A.

Theorem 2.1 highlights how "center-and-scale" normalization can inadvertently alter attention. The potential for such sign flips, demonstrated with significant likelihood, poses a serious risk to the integrity of the attention scores, as it can lead to a complete reordering of the tokens' importance.

## 2.3 ATTENTION SHIFT

Attention shift represents a critical challenge in Transformer models, directly stemming from the token shift issue. This shift perturbs the relative significance of tokens, leading to discrepancies between attention weights from normalized versus original inputs. To validate the prevalence of attention shift across normalization techniques, we conduct a study utilizing pre-trained Word2Vec embeddings Fares et al. (2017). Our analysis includes a comparison of batch normalization (BatchNorm, BN, Ioffe & Szegedy 2015), layer normalization (LayerNorm, LN, Ba et al. 2016; Vaswani et al. 2017), root mean square layer normalization (RMSNorm, RMSN, Zhang & Sennrich 2019), and our proposed unit normalization (UnitNorm, UN; see Section 3).

Our investigation utilizes sequences of token vectors, $\mathbf{X} \in \mathbb{R}^{N \times L \times D}$, as inputs to the normalization layer, where $N$ is the batch size, $L$ is the sequence length, and $D$ is the dimensionality of each token. The attention scores $\mathbf{A} \in \mathbb{R}^{N \times L \times L}$, given as Equation (3), are

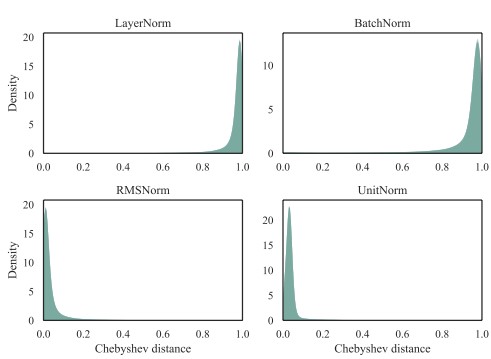 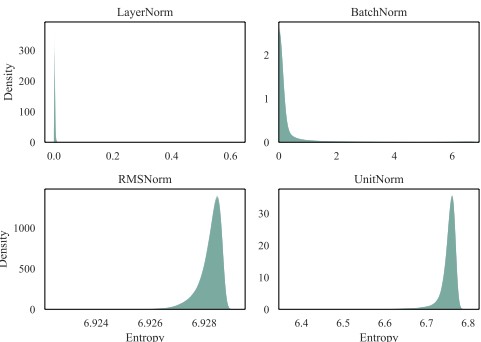

(a) Distribution of Chebyshev distance. Unit-Norm and RMSNorm preserves the distribution of attention scores; others significantly alter it.

(b) Distribution of entropy. UnitNorm and RM-SNorm preserves the high entropy of attention scores; others result in a collapsed distribution.

Figure 3: Empirical statistics for attention scores after each normalization method. Results from 10 independent experiments are overlaid. $k = 1.5$ is used for UnitNorm.

computed for 10 independent sets of 32 batches, each containing 1,024 randomly sampled embeddings from a total of 2 million. The primary goal is to assess the impact of normalization on the fidelity of attention scores $\mathbf{A}$ and $\tilde{\mathbf{A}}$, pre- and post-normalization, using the Chebyshev distance as a metric (Table S10).

$$\mathbf{A}_{n,i} = \text{softmax}\left(\frac{\mathbf{X}_{n,i}\mathbf{X}_n^T}{\sqrt{D}}\right) \tag{3}$$

where $\mathbf{A}_{n,i} \in \mathbb{R}^L$ is the attention scores for the $i$-th anchor token $\mathbf{X}_{n,i}$ to the context sequence $\mathbf{X}_n$ from the $n$-th batch; $\tilde{\mathbf{A}}$ is computed similarly from normalization output $\tilde{\mathbf{X}}$.

Chebyshev distance distributions (Figure 3(a)) reveal current methods struggle to maintain faithful attention. For them, distances cluster near 1, suggesting profound alteration of attention weights. Conversely, UnitNorm and RMSNorm demonstrates a distribution concentrated around zero, indicating minimal disruption to the original attention scores.

The empirical evidence underscores a fundamental issue with current normalization practices in Transformers: they compromise attention score fidelity, distorting relational dynamics. This harms interpretability and learning of complex dependencies. As demonstrated in our pilot study with periodic time series data (Table S2), this issue directly affects the model's ability to capture important periodic patterns that are common in time series analysis tasks.

### 2.4 SPARSE ATTENTION

The challenge of sparse attention further complicates the normalization landscape in Transformer models. Traditional "center-and-scale" methods often cause undesirable attention concentration (skewing towards single point distributions). This is due to fact that centering removes a degree of freedom from the vectors, and only query that are tightly around the $\mathbb{1}$ vector can produce uniform attention scores Brody et al. (2023). This can be depicted by the entropy of the attention scores $\mathbf{A}_i$: $H(\mathbf{A}_i) = -\sum_{j=1}^{L} \mathbf{A}_{i,j} \log \mathbf{A}_{i,j}$. A higher entropy value suggests a more uniform attention distribution, enabling models capturing periodicity in time series. Conversely, lower entropy, or a trend towards single point distributions, limits its attention to narrow ranges of tokens. While some studies Hyeon-Woo et al. (2022); Zhai et al. (2023) in other fields have shown that Transformer models may benefit from capturing longer-range, denser connections, we will show later that such sparse attention is particularly problematic in TSA tasks and requires finer control over the attention patterns.

Analysis of normalization methods through the lens of attention score entropy (Figure 3(b)) reveals a stark contrast in their effects on model behavior. BatchNorm and LayerNorm

significantly skew attention distributions towards minimal entropy. This condition not only narrows the model's focus but may also precipitate training instability Zhai et al. (2023). In contrast, UnitNorm and RMSNorm maintain higher entropy levels, suggesting a more balanced and contextually aware attention mechanism. Notably, the key deviation in attention entropy between UnitNorm and RMSNorm is the former's ability to modulate the entropy pattern by adjusting the $k$ parameter, as discussed in Section 3, while RMSNorm maintains a consistent high entropy level close to the theoretical upper bound $\log L$.

## 3 Methodology

To mitigate the challenges identified with traditional normalization methods, we introduce a novel approach called **unit normalization (UnitNorm, UN)**, formulated such that

$$\mathrm{UN}(\mathbf{X}) = D^{\frac{k}{2}} \frac{\mathbf{X}}{\|\mathbf{X}\|_2}. \tag{4}$$

UnitNorm omits centering, diverging from center-and-scale. Like RMSNorm, it scales inputs by their $\ell^2$ norm, but further scales by $D^{\frac{k}{2}}$, where $k$ controls attention sparsity.

### 3.1 Theoretical foundation

UnitNorm is theoretically grounded as a variant of LayerNorm and RMSNorm. Specifically, when taking $k = 1$, UnitNorm is effectively acting as LayerNorm with asserted zero mean, and the RMSNorm can be seen as a special case of UnitNorm with $k = 1$.

This suggests UnitNorm inherits benefits from LayerNorm and RMSNorm (e.g., mitigating vanishing/exploding gradients, stabilizing training). It ensures consistent forward/gradient propagation irrespective of learnable parameter scaling, and scales down gradients to large parameters (Proof: Section B), ensuring stability:

**Theorem 3.1** (UnitNorm preseves the gradient to the input and stablize the gradient to the learnable parameters). *Given the output of an affine transformation* $\mathbf{x} = \mathbf{W}\mathbf{v} + \mathbf{b}$, *where* $\mathbf{W}$ *and* $\mathbf{b}$ *are learnable parameters. If* $\mathbf{x}' = (\alpha\mathbf{W})\mathbf{v} + (\alpha\mathbf{b})$, *then the output of UnitNorm is unchanged, i.e.,* $\tilde{\mathbf{x}}' = \tilde{\mathbf{x}}$, *while the gradients to loss* $\mathcal{L}$ *are given as follows:*

$$\begin{aligned}
\frac{\partial \mathcal{L}}{\partial \tilde{\mathbf{x}}'} \cdot \frac{\partial \tilde{\mathbf{x}}'}{\partial (\alpha\mathbf{W})} &= \frac{1}{\alpha} \cdot \frac{\partial \mathcal{L}}{\partial \tilde{\mathbf{x}}} \cdot \frac{\partial \tilde{\mathbf{x}}}{\partial \mathbf{W}} &&= \frac{1}{\alpha} \cdot \frac{\partial \mathcal{L}}{\partial \tilde{\mathbf{x}}} \cdot \mathbf{J}\mathbf{v}^\top \\
\frac{\partial \mathcal{L}}{\partial \tilde{\mathbf{x}}'} \cdot \frac{\partial \tilde{\mathbf{x}}'}{\partial (\alpha\mathbf{b})} &= \frac{1}{\alpha} \cdot \frac{\partial \mathcal{L}}{\partial \tilde{\mathbf{x}}} \cdot \frac{\partial \tilde{\mathbf{x}}}{\partial \mathbf{b}} &&= \frac{1}{\alpha} \cdot \frac{\partial \mathcal{L}}{\partial \tilde{\mathbf{x}}} \cdot \mathbf{J} \\
\frac{\partial \mathcal{L}}{\partial \tilde{\mathbf{x}}'} \cdot \frac{\partial \tilde{\mathbf{x}}'}{\partial \mathbf{v}} &= \frac{\partial \mathcal{L}}{\partial \tilde{\mathbf{x}}} \cdot \frac{\partial \tilde{\mathbf{x}}}{\partial \mathbf{v}} &&= \frac{\partial \mathcal{L}}{\partial \tilde{\mathbf{x}}} \cdot \mathbf{J}\mathbf{W}^\top
\end{aligned} \tag{5}$$

*where* $\mathbf{J}$ *is the Jacobian matrix of* $\tilde{\mathbf{x}}$ *w.r.t.* $\mathbf{x}$.

### 3.2 Selection of $k$ values

While learnable $k$ offers flexibility, studies show fixed $k$ values often yield optimal performance Section G. Specifically, values in the range of $0.5 \sim 0.7$ have been found to be particularly effective for time series data with periodic patterns (detailed results are provided in Section G).

This optimal range can be explained by examining the entropy lower bound (ELB) characteristics. With $k \approx 0.5 \sim 0.7$, UnitNorm maintains sufficient attention diversity to capture complex patterns while still allowing for the focus on relevant tokens needed for effective periodicity recognition. This balance is critical for time series tasks where models must simultaneously recognize periodic patterns and adapt to temporal variations.

In practice, we recommend starting with $k = 0.7$ for datasets with strong periodicity, as it typically provides an excellent trade-off between sparse and dense attention distributions. For applications where the optimal $k$ value is uncertain, both fixed values ($k = 0.5, 0.7$) and learnable $k$ implementations can be evaluated to determine the best configuration.

### 3.3 Overcoming defects

By omitting centering, UnitNorm preserves input vector directions, addressing token/attention shift by maintaining dot product signs (Figure S3). It's a drop-in replacement for LayerNorm/RMSNorm in time series Transformers, needing no structural changes.

Additionally, UnitNorm confronts the sparse attention issue by introducing an entropy lower bound (ELB) for attention scores, modulated by the hyperparameter $k$ (proved in Section B). This feature enables the control of attention patterns, from dense as uniform to sparse as single point, offering versatile attention modeling:

**Theorem 3.2** (UnitNorm guarantees an entropy lower bound independent of the input). *For a given set of $L, D$ and a given $k$, there exists an entropy lower bound (ELB) of the attention scores, i.e.*

$$\mathrm{ELB}(k; L, D) = \log\left(L - 1 + e^d\right) - \frac{de^d}{L - 1 + e^d}, \tag{6}$$

*where $d = 2D^{k-\frac{1}{2}}$.*

**Corollary 3.3** (The ELB of UnitNorm can be any possible value by modulating $k$). *The ELB is a monotonically decreasing function of $k$ for a given $L, D$. Furthermore, it is bounded that $\forall k$:*

$$0 < \mathrm{ELB}(k; L, D) < \log L \tag{7}$$

The adaptability of UnitNorm is further exemplified by its applicability across variable sequence lengths, with the entropy lower bound's sensitivity to $k$ remaining relatively consistent irrespective of sequence length (Figure S4), along with the smooth landscape of $k_{50}$, the value of $k$ that achieves an ELB of $\frac{1}{2}\log L$ for a given $L, D$ pair (Figure 4), particularly with larger $D$. This property, combined with the option of setting $k$ as a learnable parameter, empowers the model to dynamically adjust its attention pattern, optimizing performance across different tasks and data sets.

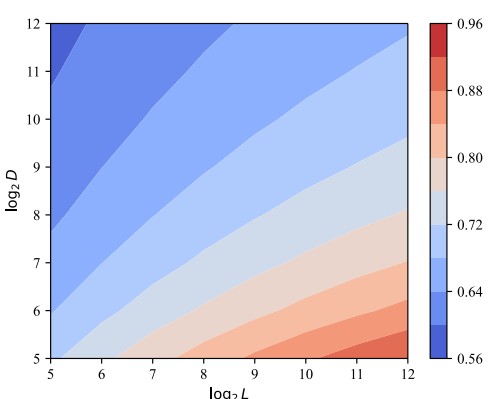

Figure 4: Landscape of $k_{50}$ for different $L, D$. The $k_{50}$ is the value of $k$ that achieves an ELB of half of the theoretical maximum $\log L$ for a given $L, D$ pair. The landscape of $k_{50}$ is rather smooth and insensitive to the sequence length $L$, indicating UnitNorm with fixed $k$ can be applied to sequences with variable length without significant change in the attention pattern.

## 4 Experiments

In our experimental evaluation, UnitNorm is rigorously tested across a spectrum of TSA tasks to illustrate its theoretical advantages in practical applications, including long term forecasting (ETTh1, ETTh2, ECL, Exchange), classification (FaceDetection, Heartbeat, PEMS-SF, UWaveGestureLibrary) and anomaly detection (MSL). We integrate UnitNorm into various Transformer models, namely Crossformer Zhang & Yan (2022), FEDformer Zhou et al. (2022), Informer Zhou et al. (2021), PatchTST Nie et al. (2022), and the vanilla Transformer Vaswani et al. (2017), all with same set of hyperparameter as described in Wu et al. (2023). For comparison, we also include BatchNorm, LayerNorm, RMSNorm, and various settings of UnitNorm (see figure legends). By doing so, we aim to demonstrate its superior ability to address normalization-related challenges, enhancing model performance in these tasks. Detailed experimental settings and full results are provided in Tables S3 to S5 and S7 to S9. Below, we outline the significance of these tasks and the specific benefits UnitNorm brings.

**Long-term forecasting**: Long-term forecasting represents a significant challenge for Transformer models, primarily due to the difficulty in maintaining periodic pattern recognition over extended sequences Li et al. (2023). The conventional normalization methods often

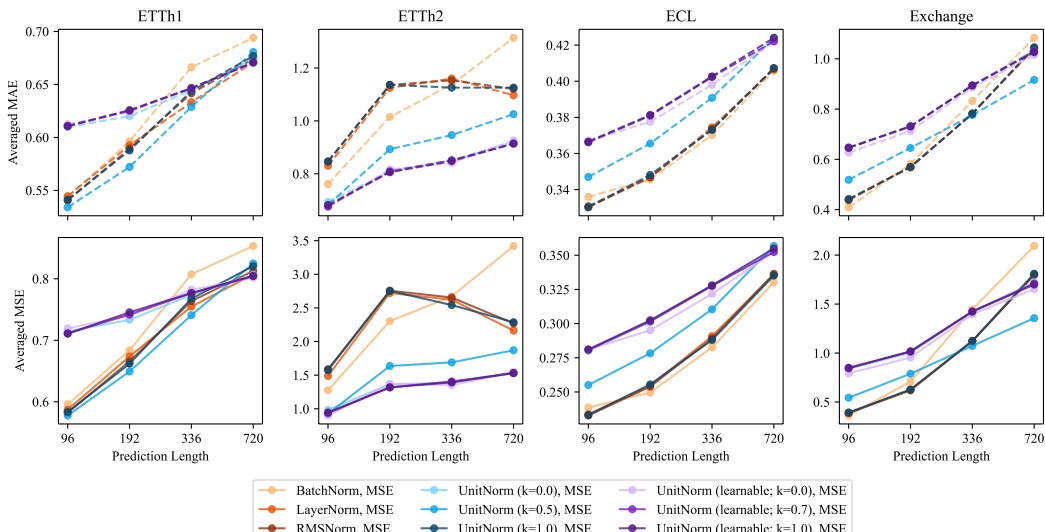

Figure 5: Average rank of normalization methods on the long-term forecasting tasks. X-axis: number of tokens to forecast, Y-axis: average rank over models. Ranks are computed based on the MAE or MSE of each model on each task with different normalization methods (lower is better). UnitNorm and UnitNorm (learnable) achieve better results with the increase of prediction horizon, and have a slower increase in prediction error compared to other normalization methods.

exacerbate the sparse attention problem, hindering the model's capability to capture periodicity. In contrast, UnitNorm excels here, with superior rank and slower error increase over longer horizons across datasets (Figure 5). With a maximum increase of 1.46/0.45 in MSE/MAE on ETTh2, and 1.27/0.36 in MSE/MAE on Exchange at the longest prediction horizon, it substantiates UnitNorm's ability to preserve the attention mechanism's effectiveness, even with increasing prediction horizons, by maintaining balanced attention and avoiding token/attention shift. Our analysis of dataset periodicity (see Appendix H.1) reveals that datasets with stronger periodic patterns, such as ETTh2, particularly benefit from UnitNorm's attention-preserving properties.

**Classification**: In classification tasks, the key challenge lies in effectively capturing long-range dependencies within sequences Vyas et al. (2022), a task at which Transformers excel. However, the efficacy of this capability can be significantly impacted by the choice of normalization method. UnitNorm, with its unique approach to normalization, has been shown to enhance model performance across multiple datasets, outperforming traditional methods in 3 out of 4 datasets on average (Figure S8), with a significant increase in accuracy of up to 4.90% on UWaveGestureLibrary, 1.95% on Heartbeat and 0.48% on FaceDetection. This underscores the versatility of UnitNorm in adapting to varied datasets, offering improved accuracy by enabling a more robust, contextually aware attention mechanism.

**Anomaly detection**: Anomaly detection in time series data demands robust model sensitivity to subtle deviations Haq & Lee (2023); Yang et al. (2023), a requirement often compromised by normalization-induced shifts in attention. The token and attention shift problems, in particular, pose significant challenges in learning stable representations. UnitNorm addresses these challenges head-on, providing a more stable foundation for anomaly detection models to operate on, therefore gaining a maximum of 7.32% in recall, 5.58% in F-score, and 2.81% in precision. Its effectiveness is dominant in all accuracy, recall, precision, and F-score metrics (Figure S9), showcasing its capacity to facilitate more accurate and reliable time series modeling for anomaly detection.

### 4.1 Extension to Large-Scale Dataset and Modern Architecture

To validate UnitNorm's generalizability beyond standard benchmarks, we also evaluated its performance on larger, more complex datasets with modern Transformer architectures.

Experiments with the Pathformer model on the Solar dataset (137 channels, 52K samples) showed that UnitNorm maintained its effectiveness even at scale, achieving the best MSE and competitive MAE scores compared to other normalization methods. This suggests that UnitNorm's benefits extend to real-world, large-scale applications and remain compatible with newer Transformer architectures. Detailed results and analysis of these experiments are provided in Appendix H.2.

## 5 DISCUSSION

**Related work** The development of effective normalization techniques is crucial in the optimization of Transformer models training Wang et al. (2019). Previous research has primarily explored two avenues: the optimal placement of normalization layers, highlighted by the Post-Layer Normalization (Post-LN) and Pre-Layer Normalization (Pre-LN) debate, which impacts training stability Xiong et al. (2020); and the normalization of model parameters, examplified by Weight Normalization Salimans & Kingma (2016). These methods aim to improve training dynamics by adjusting either the model architecture or the weights.

In contrast, our proposed UnitNorm shifts the focus from placement or parameters to the fundamental role of normalization within the attention mechanism. UnitNorm is distinguished by its emphasis on preserving information of token vectors, a core principle applicable to both Post-LN and Pre-LN configurations. This focus on normalizing layer inputs to maintain vector integrity presents a novel perspective that diverges from prior work centered on architectural adjustments or parameter optimization.

**Adopting UnitNorm in Transformer models** UnitNorm invites reconsideration of normalization practices in Transformers, suggesting alternatives that enhance model performance and stability. Its simplicity and versatility suggest it could be readily adopted across various Transformer applications. The broader impact of UnitNorm lies in its potential to improve the applicability and efficiency of Transformers in fields where precision and model stability are paramount. By addressing specific normalization-related challenges, UnitNorm can make Transformers more suitable for tasks with complex sequential relationships.

**Limitations** While UnitNorm represents a significant advancement in normalization techniques for Transformers, several areas still warrant further investigation:

- **Broader Application Scope**: Extending the application of UnitNorm beyond Transformers to other neural network architectures could provide valuable insights into the fundamental principles of normalization across deep learning models.
- **Cross Domain Validation**: Applying UnitNorm across diverse domains and challenging datasets beyond TSA, e.g., NLP Brown et al. (2020); Devlin et al. (2019) and CV Dosovitskiy et al. (2020), will further elucidate its effectiveness and generalizability, providing insights into its broad utility in deep learning.
- **Problem characterization**: Understanding how and what certain dataset characteristics influence the efficacy of normalization methods, including quantitatively assess the presence of token shift, attention shift, and sparse attention in the dynamic interplay of attention mechanisms and normalization during training, can guide the community in selecting appropriate techniques for varied deep learning challenges.

Much as UnitNorm marks a promising advancement in normalization for Transformers, its exploration is far from complete. The limitations identified herein not only highlight the need for further empirical validation across domains but also the potential for refining and extending the methodology to accommodate a wider array of architectures and applications.

**Conclusion** UnitNorm challenges prevailing normalization norms in Transformers for TSA, underscoring the need for tailored approaches. By avoiding centering, it directly addresses **token shift, attention shift, and sparse attention**, issues overlooked by traditional methods. Our contribution extends beyond the theoretical introduction of UnitNorm; it includes empirical evidence showcasing its efficacy across various tasks, setting a new precedent for normalization techniques within the Transformer architecture. UnitNorm enables more **stable and faithful representation learning**, paving the way for enhanced Transformer performance and applicability in complex sequential data analysis.

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

## A    Dimension Dependence of Sign-Flip Probability

We recall that Theorem 2.1 provided a condition for token vector means and variances, condition (1), to imply that the sign of the token dot product $\mathbf{x}^\top \mathbf{y}$ is flipped by center-and-scale standardization as in LayerNorm Ba et al. (2016).

In this section, we elucidate the dimension dependence of the required relationship between token means and standard deviations implied by this condition in the case of shared means and standard deviations across feature dimensions, such as implicitly assumed by LayerNorm.

**Corollary A.1.** *Assume that the mean and variance vectors of independent token vectors* $\mathbf{x}$ *and* $\mathbf{y}$ *satisfy* $\boldsymbol{\mu}_x = \mu_x \mathbb{1}, \boldsymbol{\mu}_y = \mu_y \mathbb{1}$ *and* $\boldsymbol{\sigma}_x^2 = \sigma_x^2 \mathbb{1}, \boldsymbol{\sigma}_y^2 = \sigma_y^2 \mathbb{1}$. *Then the mean-variance condition* (1) *of Theorem 2.1 is satisfied for all* $L \geq 77$ *if*

$$\frac{\mu_x}{\sigma_x} \geq \frac{6}{D^{1/4}} \quad and \quad \frac{\mu_y}{\sigma_y} \geq \frac{6}{D^{1/4}}, \tag{8}$$

*Furthermore, if additionally the independent token vectors are distributed as* $\mathbf{x} \sim \mathcal{N}(\boldsymbol{\mu}_x, \mathrm{diag}\left(\boldsymbol{\sigma}_x^2\right))$, $\mathbf{y} \sim \mathcal{N}(\boldsymbol{\mu}_y, \mathrm{diag}\left(\boldsymbol{\sigma}_y^2\right))$, *then the dot product* $\tilde{\mathbf{x}}^\top \tilde{\mathbf{y}}$ *of normalized vectors* $\tilde{\mathbf{x}} = \frac{\mathbf{x}-\boldsymbol{\mu}_x}{\boldsymbol{\sigma}_x}$ *and* $\tilde{\mathbf{y}} = \frac{\mathbf{y}-\boldsymbol{\mu}_y}{\boldsymbol{\sigma}_y}$ *attains a sign flip with respect to the original inner products* $\mathbf{x}^\top \mathbf{y}$ *with probability of at least 40%.*

Theorem A.1 implies that for high-dimensional token vectors with $D \gg 1$, it might become easier to satisfy (8) given an empirical token distribution, which means that sign flips of dot products after LayerNorm-style normalization might become even more prevalent in that case.

*Proof of Theorem A.1.* For the case of $\boldsymbol{\mu}_x = \mu_x \mathbb{1}, \boldsymbol{\mu}_y = \mu_y \mathbb{1}$ and $\boldsymbol{\sigma}_x^2 = \sigma_x^2 \mathbb{1}, \boldsymbol{\sigma}_y^2 = \sigma_y^2 \mathbb{1}$, it follows that

$$12 \left( \sqrt{\boldsymbol{\sigma}_x^{2\top} \boldsymbol{\sigma}_y^2} + \|\boldsymbol{\sigma}_x \circ \boldsymbol{\sigma}_y\|_\infty \right) + 5 \left( \sqrt{\boldsymbol{\sigma}_y^{2\top} \boldsymbol{\mu}_x^2} + \sqrt{\boldsymbol{\sigma}_x^{2\top} \boldsymbol{\mu}_y^2} + \|\boldsymbol{\sigma}_y \circ |\boldsymbol{\mu}_x|\|_\infty + \|\boldsymbol{\sigma}_x \circ |\boldsymbol{\mu}_y|\|_\infty \right)$$

$$= 12 \left( \sqrt{D\sigma_x^2 \sigma_y^2} + \sigma_x \sigma_y \right) + 5 \left( \sqrt{D\sigma_y^2 \mu_x^2} + \sqrt{D\sigma_x^2 \mu_y^2} + \sigma_y |\mu_x| + \sigma_x |\mu_y| \right)$$

$$\leq 12 \left( \sqrt{D \frac{D\mu_x^2 \mu_y^2}{36^2}} + \frac{D^{1/4}\mu_x}{6} \frac{D^{1/4}\mu_y}{6} \right)$$

$$+ 5 \left( \sqrt{D \frac{D^{1/2}\mu_y^2}{36} \mu_x^2} + \sqrt{D \frac{D^{1/2}\mu_x^2}{36} \mu_y^2} + \frac{D^{1/4}\mu_y}{6}|\mu_x| + \frac{D^{1/4}\mu_x}{6}|\mu_y| \right)$$

$$= 12 \left( D\frac{\mu_x \mu_y}{36} + D^{1/2}\frac{\mu_x \mu_y}{36} \right) + \frac{5}{6} \left( \sqrt{DD^{1/2}\mu_y^2 \mu_x^2} + \sqrt{DD^{1/2}\mu_x^2 \mu_y^2} + D^{1/4}\mu_y|\mu_x| + D^{1/4}\mu_x|\mu_y| \right)$$

$$= \mu_x \mu_y \left( \frac{1}{3}D + \frac{1}{3}D^{1/2} + \frac{5}{3}(D^{3/4} + D^{1/4}) \right) \leq D\mu_x \mu_y = |\boldsymbol{\mu}_x^\top \boldsymbol{\mu}_y|.$$

Here, we used in the first inequality the assumption Equation (8) and the fact that $\frac{1}{3}D^{1/2} + \frac{5}{3}(D^{3/4} + D^{1/4}) \leq \frac{2}{3}D$ for $D \geq 77$ in the last inequality. The last assertion of the theorem then follows by application of Theorem 2.1. $\qquad\square$

## B    Proofs

In this section, we detail the proofs of the theoretical results of this paper. In particular, we present the proofs of Theorem 2.1, Theorem 3.1, Theorem 3.2, Theorem 3.3, as well as of auxiliary lemmas.

### B.1    Proof of Theorem 2.1

*Proof of Theorem 2.1.* Let $\mathbf{x} \sim \mathcal{N}(\boldsymbol{\mu}_x, \mathrm{diag}\left(\boldsymbol{\sigma}_x^2\right))$ and $\mathbf{y} \sim \mathcal{N}(\boldsymbol{\mu}_y, \mathrm{diag}\left(\boldsymbol{\sigma}_y^2\right))$ be independent, and write $\mathbf{x} = (X_1, \ldots, X_D)$ and $\mathbf{y} = (Y_1, \ldots, Y_D)$, respectively.

then we can compute the expectation $\mathbb{E}\left[\mathbf{x}^\top \mathbf{y}\right]$ of the dot product of $\mathbf{x}$ and $\mathbf{y}$ as

$$
\begin{aligned}
\mathbb{E}\left[\mathbf{x}^\top \mathbf{y}\right] &= \mathbb{E}\left[\sum_{i=1}^{D} X_i Y_i\right] \\
&= \sum_{i=1}^{D} \mathbb{E}\left[X_i Y_i\right] \\
&= \sum_{i=1}^{D} \mathbb{E}\left[X_i\right]\mathbb{E}\left[Y_i\right] \\
&= \sum_{i=1}^{D} (\boldsymbol{\mu}_x)_i (\boldsymbol{\mu}_y)_i \\
&= \boldsymbol{\mu}_x^\top \boldsymbol{\mu}_y.
\end{aligned}
$$

$$
\begin{aligned}
\mathrm{Var}\left(\mathbf{x}^\top \mathbf{y}\right) &= \mathbb{E}\left[\left(\mathbf{x}^\top \mathbf{y}\right)^2\right] - \left(\mathbb{E}\left[\mathbf{x}^\top \mathbf{y}\right]\right)^2 \\
&= \mathbb{E}\left[\left(\sum_{i=1}^{D} X_i Y_i\right)^2\right] - \left(\boldsymbol{\mu}_x^\top \boldsymbol{\mu}_y\right)^2 \\
&= \sum_{i,j=1}^{D} \mathbb{E}\left[X_i Y_i X_j Y_j\right] - \left(\boldsymbol{\mu}_x^\top \boldsymbol{\mu}_y\right)^2 \\
&= \sum_{i,j=1}^{D} \mathbb{E}\left[X_i X_j\right]\mathbb{E}\left[Y_i Y_j\right] - \left(\boldsymbol{\mu}_x^\top \boldsymbol{\mu}_y\right)^2
\end{aligned}
\tag{9}
$$

By definition of covariance, we have $\boldsymbol{\sigma}_x = \mathbb{E}\left[\mathbf{x}\mathbf{x}^\top\right] - \boldsymbol{\mu}_x\boldsymbol{\mu}_x^\top$, and here $\boldsymbol{\sigma}_x = \mathrm{diag}(\boldsymbol{\sigma}_x^2)$, then Equation (9) can be simplified as follows:

$$
\begin{aligned}
\mathrm{Var}\left(\mathbf{x}^\top \mathbf{y}\right) &= \sum_{i=1}^{D} \left(\boldsymbol{\sigma}_x + \boldsymbol{\mu}_x\boldsymbol{\mu}_x^\top\right)_{ij}\left(\boldsymbol{\sigma}_y + \boldsymbol{\mu}_y\boldsymbol{\mu}_y^\top\right)_{ij} - \left(\boldsymbol{\mu}_x^\top \boldsymbol{\mu}_y\right)^2 \\
&= \langle\boldsymbol{\sigma}_x, \boldsymbol{\sigma}_y\rangle_F + \langle\boldsymbol{\mu}_x\boldsymbol{\mu}_x^\top, \boldsymbol{\sigma}_y\rangle_F + \langle\boldsymbol{\sigma}_x, \boldsymbol{\mu}_y\boldsymbol{\mu}_y^\top\rangle_F + \langle\boldsymbol{\mu}_x\boldsymbol{\mu}_x^\top, \boldsymbol{\mu}_x\boldsymbol{\mu}_x^\top\rangle_F - \left(\boldsymbol{\mu}_x^\top \boldsymbol{\mu}_y\right)^2 \\
&= \left(\boldsymbol{\sigma}_x^2\right)^\top \left(\boldsymbol{\sigma}_y^2\right) + \left(\boldsymbol{\sigma}_y^2\right)^\top \left(\boldsymbol{\mu}_x^2\right) + \left(\boldsymbol{\sigma}_x^2\right)^\top \left(\boldsymbol{\mu}_y^2\right)
\end{aligned}
\tag{10}
$$

where $\langle\cdot,\cdot\rangle_F$ is the Frobenius inner product.

Consider now the normalized random vectors $\tilde{\mathbf{x}} = \frac{\mathbf{x}-\boldsymbol{\mu}_x}{\boldsymbol{\sigma}_x}$ and $\tilde{\mathbf{y}} = \frac{\mathbf{y}-\boldsymbol{\mu}_y}{\boldsymbol{\sigma}_y}$. Due to the Gaussianity assumption on $\mathbf{x}$ and $\mathbf{y}$, it follows that the normalized vectors are also Gaussian, and in particular, are distributed as $\tilde{\mathbf{x}}, \tilde{\mathbf{y}} \sim \mathcal{N}(\mathbf{0}, \mathbb{I})$. Plugging the respective mean and variance values into the formulas for the expectation and variance for dot products above, we obtain that

$$
\mathbb{E}\left[\tilde{\mathbf{x}}^\top \tilde{\mathbf{y}}\right] = 0 \quad \text{and} \quad \mathrm{Var}\left(\tilde{\mathbf{x}}^\top \tilde{\mathbf{y}}\right) = 1 \tag{11}
$$

As $\tilde{\mathbf{x}}^\top \tilde{\mathbf{y}}$ is a symmetric random variable, it follows that

$$
\mathrm{Pr}\left(\tilde{\mathbf{x}}^\top \tilde{\mathbf{y}}\right) = 0.5. \tag{12}
$$

Next, due to the definition of the random vectors $\mathbf{x}$ and $\mathbf{y}$, it holds that $\mathbf{x}^\top \mathbf{y} = \sum_{i=1}^{D} X_i Y_i$, where $X_i \sim \mathcal{N}((\boldsymbol{\mu}_x)_i, (\boldsymbol{\sigma}_x)_i^2)$ and $Y_i \sim \mathcal{N}((\boldsymbol{\mu}_y)_i, (\boldsymbol{\sigma}_y)_i^2)$ are independent normal random variables. Going forward, we will use the $\psi_1$-Orlicz norm

$$
\|X\|_{\psi_1} := \inf\{t > 0 : \mathbb{E}[\exp(|X|/t)] \leq 2\}, \tag{13}
$$

cf. Definition 2.7.5 of Vershynin (2018). We call a random variable for which $\|\cdot\|_{\psi_1}$ is finite sub-exponential, following, e.g., Vershynin (2018).

Define now $Z_i := X_i Y_i - (\boldsymbol{\mu}_x)_i (\boldsymbol{\mu}_y)_i$. We observe that

$$Z_i = X_i Y_i - (\boldsymbol{\mu}_x)_i (\boldsymbol{\mu}_y)_i = X_i (Y_i - (\boldsymbol{\mu}_y)_i) + (X_i - (\boldsymbol{\mu}_x)_i)(\boldsymbol{\mu}_y)_i = Z_i^{(1)} + Z_i^{(2)}$$

with $Z_i^{(1)} := X_i (Y_i - (\boldsymbol{\mu}_y)_i)$ and $Z_i^{(2)} := (X_i - (\boldsymbol{\mu}_x)_i)(\boldsymbol{\mu}_y)_i$. To bound the $\psi_1$-norm of $Z_i$, we bound this norm for $Z_i^{(1)}$ and $Z_i^{(2)}$ separately.

Indeed, due to Lemma 2.7.7 of Vershynin (2018), it holds that

$$\|Z_i^{(1)}\|_{\psi_1} \leq \|X_i\|_{\psi_2} \|Y_i - (\boldsymbol{\mu}_y)_i\|_{\psi_2},$$

where

$$\|X\|_{\psi_2} := \inf\{t > 0 : \mathbb{E}[\exp(X^2/t^2)] \leq 2\}, \tag{14}$$

is the $\psi_2$-Orlicz norm Vershynin (2018) characterizing sub-Gaussian random variables $X$. From Lemma B.2, it follows therefore that

$$\|Z_i^{(1)}\|_{\psi_1} \leq \max\left(2(\boldsymbol{\sigma}_x)_i, \sqrt{\frac{(\boldsymbol{\mu}_x)_i^2}{\log 2} + (\boldsymbol{\sigma}_x)_i^2}\right) \sqrt{\frac{8}{3}}(\boldsymbol{\sigma}_y)_i.$$

For the second part, since $\|\cdot\|_{\psi_2}$ is a norm, we estimate that

$$\|Z_i^{(2)}\|_{\psi_1} \leq \|X_i - (\boldsymbol{\mu}_x)_i\|_{\psi_2} \|(\boldsymbol{\mu}_y)_i\|_{\psi_2} \leq \sqrt{\frac{8}{3}}(\boldsymbol{\sigma}_x)_i \|(\boldsymbol{\mu}_y)_i\|_{\psi_2} \leq \sqrt{\frac{8}{3}}(\boldsymbol{\sigma}_x)_i \frac{|(\boldsymbol{\mu}_y)_i|}{\sqrt{\log 2}},$$

where we used again Lemma 2.7.7 and (2.17) of Vershynin (2018) in the first and last inequality, respectively, and Lemma B.2 in the second inequality.

From this, it follows that

$$\|Z_i\|_{\psi_1} \leq \|Z_i^{(1)}\|_{\psi_1} + \|Z_i^{(2)}\|_{\psi_1} \leq \max\left(2(\boldsymbol{\sigma}_x)_i, \sqrt{\frac{(\boldsymbol{\mu}_x)_i^2}{\log 2} + (\boldsymbol{\sigma}_x)_i^2}\right) \sqrt{\frac{8}{3}}(\boldsymbol{\sigma}_y)_i + \sqrt{\frac{8}{3}}(\boldsymbol{\sigma}_x)_i \frac{|(\boldsymbol{\mu}_y)_i|}{\sqrt{\log 2}}$$

$$\leq \left(2(\boldsymbol{\sigma}_x)_i + \sqrt{\frac{(\boldsymbol{\mu}_x)_i^2}{\log 2} + (\boldsymbol{\sigma}_x)_i^2}\right) \sqrt{\frac{8}{3}}(\boldsymbol{\sigma}_y)_i + \sqrt{\frac{8}{3}}(\boldsymbol{\sigma}_x)_i \frac{|(\boldsymbol{\mu}_y)_i|}{\sqrt{\log 2}}$$

$$\leq 2\sqrt{6}(\boldsymbol{\sigma}_x)_i(\boldsymbol{\sigma}_y)_i + \sqrt{\frac{8}{3}}(\boldsymbol{\sigma}_y)_i \frac{|(\boldsymbol{\mu}_x)_i|}{\sqrt{\log 2}} + \sqrt{\frac{8}{3}}(\boldsymbol{\sigma}_x)_i \frac{|(\boldsymbol{\mu}_y)_i|}{\sqrt{\log 2}},$$

using that $\sqrt{a^2 + b^2} \leq a + b$ for any non-negative $a, b \geq 0$ in the last inequality. We next establish a lower bound on the probability of a sign flip through normalization, i.e., for $\Pr(\text{sgn}(\mathbf{x}^\top \mathbf{y}) \neq \text{sgn}(\tilde{\mathbf{x}}^\top \tilde{\mathbf{y}}))$. Assuming without loss of generality that $|\boldsymbol{\mu}_x^\top \boldsymbol{\mu}_y| = \boldsymbol{\mu}_x^\top \boldsymbol{\mu}_y$, we observe that

$$\Pr\left(\text{sgn}(\mathbf{x}^\top \mathbf{y}) \neq \text{sgn}(\tilde{\mathbf{x}}^\top \tilde{\mathbf{y}})\right) = \Pr\left((\mathbf{x}^\top \mathbf{y} > 0) \wedge (\tilde{\mathbf{x}}^\top \tilde{\mathbf{y}} < 0)\right) + \Pr\left((\mathbf{x}^\top \mathbf{y} < 0) \wedge (\tilde{\mathbf{x}}^\top \tilde{\mathbf{y}} > 0)\right)$$

$$\geq \Pr\left((\mathbf{x}^\top \mathbf{y} > 0) \wedge (\tilde{\mathbf{x}}^\top \tilde{\mathbf{y}} < 0)\right).$$

Furthermore, since the distribution of the normalized vectors $\tilde{\mathbf{x}}$ and $\tilde{\mathbf{y}}$ is symmetric, the same holds true for the dot product $\tilde{\mathbf{x}}^\top \tilde{\mathbf{y}}$, which implies that

$$\Pr\left((\mathbf{x}^\top \mathbf{y} > 0) \wedge (\tilde{\mathbf{x}}^\top \tilde{\mathbf{y}} < 0)\right) = 1 - \Pr\left((\mathbf{x}^\top \mathbf{y} \leq 0) \vee (\tilde{\mathbf{x}}^\top \tilde{\mathbf{y}} \geq 0)\right) \geq 1 - \Pr\left(\mathbf{x}^\top \mathbf{y} \leq 0\right) - \Pr\left(\tilde{\mathbf{x}}^\top \tilde{\mathbf{y}} \geq 0\right)$$

$$\geq 1 - 0.5 - \Pr\left(\mathbf{x}^\top \mathbf{y} \leq 0\right) = 0.5 - \Pr\left(\mathbf{x}^\top \mathbf{y} \leq 0\right).$$

It remains to show that

$$\Pr\left(\mathbf{x}^\top \mathbf{y} \leq 0\right) \leq 0.1. \tag{16}$$

To establish this, we see that

$$\Pr\left(\mathbf{x}^\top \mathbf{y} \leq 0\right) = \Pr\left(\mathbf{x}^\top \mathbf{y} - \mathbb{E}\left[\mathbf{x}^\top \mathbf{y}\right] \leq -\mathbb{E}\left[\mathbf{x}^\top \mathbf{y}\right]\right) = \Pr\left(\mathbf{x}^\top \mathbf{y} - \boldsymbol{\mu}_x^\top \boldsymbol{\mu}_y \leq -\boldsymbol{\mu}_x^\top \boldsymbol{\mu}_y\right)$$

$$= \Pr\left(\sum_{i=1}^{D} Z_i \leq -\boldsymbol{\mu}_x^\top \boldsymbol{\mu}_y\right)$$

with the random variables $Z_i$ defined above. Using the triangle inequality of the $\ell_2$-norm, it follows from (15) that

$$\sqrt{\sum_{i=1}^{D}\|Z_i\|_{\psi_1}^2} \leq 2\sqrt{6}\sqrt{(\boldsymbol{\sigma}_x^2)^\top\boldsymbol{\sigma}_y^2} + \sqrt{\frac{8}{3\log 2}}\left(\sqrt{(\boldsymbol{\sigma}_y^2)^\top\boldsymbol{\mu}_x^2} + \sqrt{(\boldsymbol{\sigma}_x^2)^\top\boldsymbol{\mu}_y^2}\right)$$

and that

$$\max_{i=1}^{D}\|Z_i\|_{\psi_1} \leq 2\sqrt{6}\|\boldsymbol{\sigma}_x \circ \boldsymbol{\sigma}_y\|_\infty + \sqrt{\frac{8}{3\log 2}}\left(\|\boldsymbol{\sigma}_y \circ |(\boldsymbol{\mu}_x)|\|_\infty + \|\boldsymbol{\sigma}_x \circ |(\boldsymbol{\mu}_y)|\|_\infty\right),$$

which implies that

$$\sqrt{2\sum_{i=1}^{D}\|Z_i\|_{\psi_1}^2}\sqrt{\log(10)} + \max_{i=1}^{D}\|Z_i\|_{\psi_1}\log(10)$$

$$\leq 4\sqrt{3\log(10)}\sqrt{(\boldsymbol{\sigma}_x^2)^\top\boldsymbol{\sigma}_y^2} + \frac{4\sqrt{\log(10)}}{\sqrt{3\log(2)}}\left(\sqrt{(\boldsymbol{\sigma}_y^2)^\top\boldsymbol{\mu}_x^2} + \sqrt{(\boldsymbol{\sigma}_x^2)^\top\boldsymbol{\mu}_y^2}\right)$$

$$+ 2\sqrt{6}\log(10)\|\boldsymbol{\sigma}_x \circ \boldsymbol{\sigma}_y\|_\infty + \sqrt{\frac{8}{3\log 2}}\log(10)\left(\|\boldsymbol{\sigma}_y \circ |(\boldsymbol{\mu}_x)|\|_\infty + \|\boldsymbol{\sigma}_x \circ |(\boldsymbol{\mu}_y)|\|_\infty\right)$$

$$\leq 12\left(\sqrt{(\boldsymbol{\sigma}_x^2)^\top\boldsymbol{\sigma}_y^2} + \|\boldsymbol{\sigma}_x \circ \boldsymbol{\sigma}_y\|_\infty\right) + 5\left(\sqrt{(\boldsymbol{\sigma}_y^2)^\top\boldsymbol{\mu}_x^2} + \sqrt{(\boldsymbol{\sigma}_x^2)^\top\boldsymbol{\mu}_y^2} + \|\boldsymbol{\sigma}_y \circ |(\boldsymbol{\mu}_x)|\|_\infty + \|\boldsymbol{\sigma}_x \circ |(\boldsymbol{\mu}_y)|\|_\infty\right)$$

$$\leq |\boldsymbol{\mu}_x^\top\boldsymbol{\mu}_y|,$$

using the assumption (1) in the last inequality. With this inequality, we can use the fact that the $Z_i$ are independent mean-zero sub-exponential random variables and Bernstein's inequality as stated in Lemma B.1 to conclude that

$$\Pr\left(\sum_{i=1}^{D}Z_i \leq -\boldsymbol{\mu}_x^\top\boldsymbol{\mu}_y\right) \leq \Pr\left(\sum_{i=1}^{D}Z_i \leq -\left(\sqrt{2\sum_{i=1}^{D}\|Z_i\|_{\psi_1}^2}\sqrt{\log(10)} + \max_{i=1}^{D}\|Z_i\|_{\psi_1}\log(10)\right)\right)$$

$$\leq \exp(-\log(10)) = 0.1.$$

This establishes (16), which concludes the proof. $\qquad\square$

**Lemma B.1** (Bernstein's Inequality, cf. Lemma 5.1 of Dirksen (2015)). *Let $Z_1, \ldots Z_D$ be independent mean-zero sub-exponential random variables. Then for every $t \geq 0$,*

$$\Pr\left(\sum_{i=1}^{D}Z_i \leq -\left(\sqrt{2\sum_{i=1}^{D}\|Z_i\|_{\psi_1}^2}\sqrt{t} + \max_{i=1}^{D}\|Z_i\|_{\psi_1}t\right)\right) \leq \exp(-t).$$

**Lemma B.2** (Bounds on $\psi_2$-norm of Gaussians Vershynin (2018)). *1. If $X \sim \mathcal{N}(0,\sigma^2)$ is a centered Gaussian random variable with variance $\sigma^2$, then its $\psi_2$-norm (14) satisfies*

$$\|X\|_{\psi_2} \leq \sqrt{\frac{8}{3}}\sigma.$$

*2. If $X \sim \mathcal{N}(\mu,\sigma^2)$ is a Gaussian random variable with mean $\mu$ and variance $\sigma^2$, then its $\psi_2$-norm (14) satisfies*

$$\|X\|_{\psi_2} \leq \max\left(2\sigma, \sqrt{\frac{\mu^2}{\log 2} + \sigma^2}\right).$$

## B.2 Proof of Theorem 3.1

*Proof of Theorem 3.1.* Given the output of an affine transformation $\mathbf{x} = \mathbf{W}\mathbf{v} + \mathbf{b}$, where $\mathbf{W}$ and $\mathbf{b}$ are learnable parameters. If $\mathbf{x}' = (\alpha\mathbf{W})\mathbf{v} + (\alpha\mathbf{b})$, then the output of UnitNorm is unchanged, *i.e.*, $\tilde{\mathbf{x}}' = \tilde{\mathbf{x}}$, while the gradients to loss $\mathcal{L}$ are given as follows:

$$
\begin{aligned}
\frac{\partial \mathcal{L}}{\partial \tilde{\mathbf{x}}'} \cdot \frac{\partial \tilde{\mathbf{x}}'}{\partial (\alpha\mathbf{W})} &= \frac{1}{\alpha} \cdot \frac{\partial \mathcal{L}}{\partial \tilde{\mathbf{x}}} \cdot \frac{\partial \tilde{\mathbf{x}}}{\partial \mathbf{W}} &&= \frac{1}{\alpha} \cdot \frac{\partial \mathcal{L}}{\partial \tilde{\mathbf{x}}} \cdot \mathbf{J}\mathbf{v}^\top \\
\frac{\partial \mathcal{L}}{\partial \tilde{\mathbf{x}}'} \cdot \frac{\partial \tilde{\mathbf{x}}'}{\partial (\alpha\mathbf{b})} &= \frac{1}{\alpha} \cdot \frac{\partial \mathcal{L}}{\partial \tilde{\mathbf{x}}} \cdot \frac{\partial \tilde{\mathbf{x}}}{\partial \mathbf{b}} &&= \frac{1}{\alpha} \cdot \frac{\partial \mathcal{L}}{\partial \tilde{\mathbf{x}}} \cdot \mathbf{J} \\
\frac{\partial \mathcal{L}}{\partial \tilde{\mathbf{x}}'} \cdot \frac{\partial \tilde{\mathbf{x}}'}{\partial \mathbf{v}} &= \frac{\partial \mathcal{L}}{\partial \tilde{\mathbf{x}}} \cdot \frac{\partial \tilde{\mathbf{x}}}{\partial \mathbf{v}} &&= \frac{\partial \mathcal{L}}{\partial \tilde{\mathbf{x}}} \cdot \mathbf{J}\mathbf{W}^\top
\end{aligned}
\tag{17}
$$

Proof: First we will show $\tilde{\mathbf{x}}' = \tilde{\mathbf{x}}$, for which we have:

$$
\begin{aligned}
\tilde{\mathbf{x}}' &= D^{\frac{k}{2}} \frac{\mathbf{x}'}{\|\mathbf{x}'\|} \\
&= D^{\frac{k}{2}} \frac{\alpha\mathbf{x}}{\alpha \|\mathbf{x}\|} \\
&= D^{\frac{k}{2}} \frac{\mathbf{x}}{\|\mathbf{x}\|} \\
&= \tilde{\mathbf{x}}
\end{aligned}
\tag{18}
$$

And thus for the gradients to loss $\mathcal{L}$, we have $\frac{\partial \mathcal{L}}{\partial \tilde{\mathbf{x}}'} = \frac{\partial \mathcal{L}}{\partial \tilde{\mathbf{x}}}$. Also, for the Jacobian matrix $\mathbf{J}$ of $\tilde{\mathbf{x}}$ *w.r.t.* $\mathbf{x}$, we have

$$
\begin{aligned}
\mathbf{J} &= \frac{\partial D^{\frac{k}{2}} \frac{\mathbf{x}}{\|\mathbf{x}\|}}{\partial \mathbf{x}} \\
&= D^{\frac{k}{2}} \left( \frac{\mathbf{I}}{\|\mathbf{x}\|} - \frac{\mathbf{x}\mathbf{x}^\top}{\|\mathbf{x}\|^3} \right)
\end{aligned}
\tag{19}
$$

And the Jacobian matrix of $\tilde{\mathbf{x}}'$ *w.r.t.* $\mathbf{x}'$ is given as:

$$
\begin{aligned}
\frac{\partial D^{\frac{k}{2}} \frac{\mathbf{x}'}{\|\mathbf{x}'\|}}{\partial \mathbf{x}'} &= D^{\frac{k}{2}} \left( \frac{\mathbf{I}}{\|\mathbf{x}'\|} - \frac{\mathbf{x}'\mathbf{x}'^\top}{\|\mathbf{x}'\|^3} \right) \\
&= D^{\frac{k}{2}} \left( \frac{\mathbf{I}}{\alpha \|\mathbf{x}\|} - \frac{\alpha^2 \mathbf{x}\mathbf{x}^\top}{\alpha^3 \|\mathbf{x}\|^3} \right) \\
&= \frac{1}{\alpha} D^{\frac{k}{2}} \left( \frac{\mathbf{I}}{\|\mathbf{x}\|} - \frac{\mathbf{x}\mathbf{x}^\top}{\|\mathbf{x}\|^3} \right) \\
&= \frac{1}{\alpha} \mathbf{J}
\end{aligned}
\tag{20}
$$

Then we have the gradient of loss *w.r.t.* $\mathbf{W}$ and $\alpha\mathbf{W}$:

$$
\begin{aligned}
\frac{\partial \mathcal{L}}{\partial \tilde{\mathbf{x}}} \cdot \frac{\partial \tilde{\mathbf{x}}}{\partial \mathbf{W}} &= \frac{\partial \mathcal{L}}{\partial \tilde{\mathbf{x}}} \cdot \frac{\partial \tilde{\mathbf{x}}}{\partial \mathbf{x}} \cdot \frac{\partial \mathbf{x}}{\partial \mathbf{W}} \\
&= \frac{\partial \mathcal{L}}{\partial \tilde{\mathbf{x}}} \cdot \mathbf{J}\mathbf{v}^\top \\
\frac{\partial \mathcal{L}}{\partial \tilde{\mathbf{x}}'} \cdot \frac{\partial \tilde{\mathbf{x}}'}{\partial (\alpha\mathbf{W})} &= \frac{\partial \mathcal{L}}{\partial \tilde{\mathbf{x}}'} \cdot \frac{\partial \tilde{\mathbf{x}}'}{\partial \mathbf{x}'} \cdot \frac{\partial \mathbf{x}'}{\partial (\alpha\mathbf{W})} \\
&= \frac{\partial \mathcal{L}}{\partial \tilde{\mathbf{x}}} \cdot \frac{1}{\alpha} \mathbf{J}\mathbf{v}^\top \\
\Rightarrow \frac{\partial \mathcal{L}}{\partial \tilde{\mathbf{x}}'} \cdot \frac{\partial \tilde{\mathbf{x}}'}{\partial (\alpha\mathbf{W})} &= \frac{1}{\alpha} \cdot \frac{\partial \mathcal{L}}{\partial \tilde{\mathbf{x}}} \cdot \frac{\partial \tilde{\mathbf{x}}}{\partial \mathbf{W}} = \frac{1}{\alpha} \cdot \frac{\partial \mathcal{L}}{\partial \tilde{\mathbf{x}}} \cdot \mathbf{J}\mathbf{v}^\top
\end{aligned}
\tag{21}
$$

Similarly, for $\mathbf{b}$ and $\alpha\mathbf{b}$ we have:

$$
\begin{aligned}
\frac{\partial \mathcal{L}}{\partial \tilde{\mathbf{x}}} \cdot \frac{\partial \tilde{\mathbf{x}}}{\partial \mathbf{b}} &= \frac{\partial \mathcal{L}}{\partial \tilde{\mathbf{x}}} \cdot \frac{\partial \tilde{\mathbf{x}}}{\partial \mathbf{x}} \cdot \frac{\partial \mathbf{x}}{\partial \mathbf{b}} \\
&= \frac{\partial \mathcal{L}}{\partial \tilde{\mathbf{x}}} \cdot \mathbf{J} \\
\frac{\partial \mathcal{L}}{\partial \tilde{\mathbf{x}}'} \cdot \frac{\partial \tilde{\mathbf{x}}'}{\partial (\alpha\mathbf{b})} &= \frac{\partial \mathcal{L}}{\partial \tilde{\mathbf{x}}'} \cdot \frac{\partial \tilde{\mathbf{x}}'}{\partial \mathbf{x}'} \cdot \frac{\partial \mathbf{x}'}{\partial (\alpha\mathbf{b})} \\
&= \frac{\partial \mathcal{L}}{\partial \tilde{\mathbf{x}}} \cdot \frac{1}{\alpha}\mathbf{J} \\
\Rightarrow \frac{\partial \mathcal{L}}{\partial \tilde{\mathbf{x}}'} \cdot \frac{\partial \tilde{\mathbf{x}}'}{\partial (\alpha\mathbf{b})} &= \frac{1}{\alpha} \cdot \frac{\partial \mathcal{L}}{\partial \tilde{\mathbf{x}}} \cdot \frac{\partial \tilde{\mathbf{x}}}{\partial \mathbf{b}} = \frac{1}{\alpha} \cdot \frac{\partial \mathcal{L}}{\partial \tilde{\mathbf{x}}} \cdot \mathbf{J}
\end{aligned}
\tag{22}
$$

And for $\mathbf{v}$, we have:

$$
\begin{aligned}
\frac{\partial \mathcal{L}}{\partial \tilde{\mathbf{x}}} \cdot \frac{\partial \tilde{\mathbf{x}}}{\partial \mathbf{v}} &= \frac{\partial \mathcal{L}}{\partial \tilde{\mathbf{x}}} \cdot \frac{\partial \tilde{\mathbf{x}}}{\partial \mathbf{x}} \cdot \frac{\partial \mathbf{x}}{\partial \mathbf{v}} \\
&= \frac{\partial \mathcal{L}}{\partial \tilde{\mathbf{x}}} \cdot \mathbf{J}\mathbf{W}^\top \\
\frac{\partial \mathcal{L}}{\partial \tilde{\mathbf{x}}'} \cdot \frac{\partial \tilde{\mathbf{x}}'}{\partial \mathbf{v}} &= \frac{\partial \mathcal{L}}{\partial \tilde{\mathbf{x}}'} \cdot \frac{\partial \tilde{\mathbf{x}}'}{\partial \mathbf{x}'} \cdot \frac{\partial \mathbf{x}'}{\partial \mathbf{v}} \\
&= \frac{\partial \mathcal{L}}{\partial \tilde{\mathbf{x}}} \cdot \frac{1}{\alpha}\mathbf{J}(\alpha\mathbf{W})^\top \\
\Rightarrow \frac{\partial \mathcal{L}}{\partial \tilde{\mathbf{x}}'} \cdot \frac{\partial \tilde{\mathbf{x}}'}{\partial \mathbf{v}} &= \frac{\partial \mathcal{L}}{\partial \tilde{\mathbf{x}}} \cdot \frac{\partial \tilde{\mathbf{x}}}{\partial \mathbf{v}} = \frac{\partial \mathcal{L}}{\partial \tilde{\mathbf{x}}} \cdot \mathbf{J}\mathbf{W}^\top
\end{aligned}
\tag{23}
$$

$\square$

### B.3 Proofs of Theorem 3.2 and Theorem 3.3

*Proof of Theorem 3.2.* Let $\mathbf{X} \in \mathbb{R}^{L \times D}$ be a single sequence of token vectors, and let $\tilde{\mathbf{X}}$ be the unit normalized output with modulus $k$, the entropy lower bound (ELB) of the attention scores is given by the following expression:

$$
\begin{aligned}
\mathrm{ELB}(k; L, D) &= \min_{i=1}^{L} H(\mathbf{A}_i) \\
&= \min_{i=1}^{L} \left( -\sum_{j=1}^{L} \mathbf{A}_{i,j} \log \mathbf{A}_{i,j} \right) \\
&= \log\left( L - 1 + \exp\left( 2D^{k-\frac{1}{2}} \right) \right) - \frac{2D^{k-\frac{1}{2}} \exp\left( 2D^{k-\frac{1}{2}} \right)}{L - 1 + \exp\left( 2D^{k-\frac{1}{2}} \right)}
\end{aligned}
\tag{24}
$$

Proof: Let $\tilde{\mathbf{X}} = D^{\frac{k}{2}} \mathbf{e}$ where $\mathbf{e}$ are the vectors of unit norm. Without loss of generality, we can assume the ELB is achieved at anchor index $i$, where we can compute the attention scores as follows:

$$
\begin{aligned}
\mathbf{A}_i &= \mathrm{softmax}\left( \frac{\tilde{\mathbf{X}}_i \tilde{\mathbf{X}}^\top}{\sqrt{D}} \right) \\
&= \mathrm{softmax}\left( \frac{D^k \mathbf{e}_i \mathbf{e}^\top}{\sqrt{D}} \right) \\
&= \mathrm{softmax}\left( D^{k-\frac{1}{2}} \mathbf{e}_i \mathbf{e}^\top \right)
\end{aligned}
\tag{25}
$$

Since $\mathbf{e}_i \mathbf{e}_j^\top \in (-1, 1)$, $\forall i, j = 1, 2, \cdots, L$, the entropy of the attentions scores is lower bounded by the following expression when it satisfies that $\mathbf{e}_i \mathbf{e}_j = \begin{cases} 1, & j = i \\ -1, & j \neq i \end{cases}$:

$$
\begin{aligned}
&H(\mathbf{A}_i) \\
&= -\sum_{j=1}^{L} \mathbf{A}_{i,j} \log \mathbf{A}_{i,j} \\
&= -(L-1) \cdot \frac{\exp\left(-D^{k-\frac{1}{2}}\right)}{(L-1)\exp\left(-D^{k-\frac{1}{2}}\right) + \exp\left(D^{k-\frac{1}{2}}\right)} \log \frac{\exp\left(-D^{k-\frac{1}{2}}\right)}{(L-1)\exp\left(-D^{k-\frac{1}{2}}\right) + \exp\left(D^{k-\frac{1}{2}}\right)} \\
&\quad - \frac{\exp\left(D^{k-\frac{1}{2}}\right)}{(L-1)\exp\left(-D^{k-\frac{1}{2}}\right) + \exp\left(D^{k-\frac{1}{2}}\right)} \log \frac{\exp\left(D^{k-\frac{1}{2}}\right)}{(L-1)\exp\left(-D^{k-\frac{1}{2}}\right) + \exp\left(D^{k-\frac{1}{2}}\right)} \\
&= \frac{L-1}{L-1+\exp\left(2D^{k-\frac{1}{2}}\right)} \log\left(L-1+\exp\left(2D^{k-\frac{1}{2}}\right)\right) \\
&\quad + \frac{\exp\left(2D^{k-\frac{1}{2}}\right)}{L-1+\exp\left(2D^{k-\frac{1}{2}}\right)} \log \frac{L-1+\exp\left(2D^{k-\frac{1}{2}}\right)}{\exp\left(2D^{k-\frac{1}{2}}\right)} \\
&= \log\left(L-1+\exp\left(2D^{k-\frac{1}{2}}\right)\right) - \frac{2D^{k-\frac{1}{2}}\exp\left(2D^{k-\frac{1}{2}}\right)}{L-1+\exp\left(2D^{k-\frac{1}{2}}\right)}
\end{aligned}
$$

(26)

Therefore, the entropy lower bound (ELB) for any $L, D$ and $k$ is:

$$
\mathrm{ELB}(k; L, D) = \log\left(L-1+\exp\left(2D^{k-\frac{1}{2}}\right)\right) - \frac{2D^{k-\frac{1}{2}}\exp\left(2D^{k-\frac{1}{2}}\right)}{L-1+\exp\left(2D^{k-\frac{1}{2}}\right)}
\tag{27}
$$

$\square$

*Proof of Theorem 3.3.* The ELB is a monotonically decreasing function of $k$ bounded between $0$ and $\log L$.

Proof: Let $d = 2D^{k-\frac{1}{2}}$, then it is obvious that $d$ is monotonically increasing with $k$, therefore we only need to prove that $\mathrm{ELB}(k; L, D)$ is monotonically decreasing with $d$. The derivative of $\mathrm{ELB}(k; L, D)$ with respect to $d$ is given as follows:

$$
\begin{aligned}
\frac{\partial\,\mathrm{ELB}(k; L, D)}{\partial d} &= \frac{e^d}{L-1+e^d} - \frac{\left(L-1+e^d\right)(d+1)e^d - (de^d)e^d}{\left(L-1+e^d\right)^2} \\
&= \frac{e^d}{\left(L-1+e^d\right)^2}\left(\left(L-1+e^d\right) - \left(L-1+e^d\right)(d+1) + de^d\right) \\
&= \frac{de^d}{\left(L-1+e^d\right)^2}(1-L) \\
&\quad (\forall L > 1) < 0
\end{aligned}
\tag{28}
$$

Therefore, $\mathrm{ELB}(k; L, D)$ is monotonically decreasing with $d$ and with $k$. If the limits of $\mathrm{ELB}(k; L, D)$ as $k \to -\infty$ and $k \to +\infty$ exist, then $\mathrm{ELB}(k; L, D)$ is bounded between these

two limits. The limits are given as follows:

$$
\begin{aligned}
\lim_{k \to -\infty} \mathrm{ELB}(k; L, D) &= \lim_{d \to 0^+} \left( \log \left( L - 1 + e^d \right) - \frac{d e^d}{L - 1 + e^d} \right) \\
&= \log \left( L - 1 + 1 \right) - \frac{0}{L - 1 + 1} \\
&= \log L
\end{aligned}
\tag{29}
$$

$$
\begin{aligned}
\lim_{k \to +\infty} \mathrm{ELB}(k; L, D) &= \lim_{d \to +\infty} \left( \log \left( L - 1 + e^d \right) - \frac{d e^d}{L - 1 + e^d} \right) \\
&= \lim_{d \to +\infty} \log e^d - \lim_{d \to +\infty} \frac{d}{(L-1)e^{-d} + 1} \\
&= d - d \\
&= 0
\end{aligned}
\tag{30}
$$

Therefore, $\mathrm{ELB}(k; L, D)$ is bounded between $0$ and $\log L$. $\qquad \square$

## C  Discussion

### C.1  Difference between the proposed normalization and the other normalization

BatchNorm and LayerNorm are all normalization methods that are widely used in deep learning. They share the same center-and-scale normalization paradigm by first subtracting the mean and then divide by standard deviation. The only difference between them in terms of computation is the dimensions of data used to compute these statistics, as shown in Table S1.

In terms of application, BatchNorm is often used in fully connected layers and convolution layers, while LayerNorm is often used in recurrent neural networks and Transformers. The subtle difference between LayerNorm (theory) and LayerNorm (practice) might be attributed to the fact that the sequence length $L$ is often variable in Transformers, thus normalization within each token might be more stable. But this will require further investigation to come to a conclusion.

The proposed UnitNorm is a normalization method that is used to normalize the input data to have unit norm, which takes the same dimension for computation as LayerNorm, yet it distinguishes itself from LayerNorm by the fact that it does not subtract the mean and divide by standard deviation. Also, UnitNorm discard the center operation on the normalized output, as it will also cause the problem of token shift (Section 2.2).

### C.2  Feasibility of switching the order of normalization and projection in theoretical analysis

Let $\mathbf{X} \in \mathbb{R}^{L \times D}$ be a single sequence of token vectors, and the normalization operation is given in the following form:

$$
f : \mathbf{X} \mapsto \frac{\mathbf{X} - \boldsymbol{\mu}}{\boldsymbol{\sigma}} \equiv \mathbf{X}\mathbf{W} + \mathbf{b}
\tag{31}
$$

where $\boldsymbol{\mu}$ and $\boldsymbol{\sigma}$ are the mean and standard deviation of the input vector $\mathbf{X}$, respectively, and $\mathbf{W} = \boldsymbol{\sigma}^{-1}$ and $\mathbf{b} = \boldsymbol{\mu}\boldsymbol{\sigma}^{-1}$. Depending on the normalization method, the mean and standard deviation can be computed over different dimensions.

The projection in the attention mechanism maps the input vectors to query, key and value vectors, and here we only consider the query and key vectors for this discussion, which are computed as follows:

$$
\begin{aligned}
\mathbf{Q} &= \mathbf{X}\mathbf{W}_Q + \mathbf{b}_Q \\
\mathbf{K} &= \mathbf{X}\mathbf{W}_K + \mathbf{b}_K
\end{aligned}
\tag{32}
$$

Table S1: Computation of the statistics for different normalization methods. Input data $\mathbf{X} \in \mathbb{R}^{N \times L \times D}$, where $N$ is the batch size, $L$ is the sequence length and $D$ is the feature dimension. $\mathbf{X}_{n,l,d}$ denotes the $d$-th feature of the $l$-th token in the $n$-th sequence. Normalization is broadcasted over the same dimension as the statistics and mathematical operations are done element-wise. For BatchNorm, LayerNorm (theory) and LayerNorm (practice), $\boldsymbol{\gamma}$ and $\boldsymbol{\beta}$ are optional learnable parameters that will re-scale and re-center the normalized output element-wise, which is enabled by default in the PyTorch's implementation.

| Method | Statistics | Normalization |
|---|---|---|
| BatchNorm | $\mu_d = \dfrac{1}{NL} \sum_{n=1}^{N} \sum_{l=1}^{L} \mathbf{X}_{n,l,d}$ $\sigma_d^2 = \dfrac{1}{NL} \sum_{n=1}^{N} \sum_{l=1}^{L} (\mathbf{X}_{n,l,d} - \mu_d)^2$ $\boldsymbol{\mu} = [\mu_1 \quad \mu_2 \quad \cdots \quad \mu_D]^\top \in \mathbb{R}^{1 \times 1 \times D}$ $\boldsymbol{\sigma}^2 = [\sigma_1^2 \quad \sigma_2^2 \quad \cdots \quad \sigma_D^2]^\top \in \mathbb{R}^{1 \times 1 \times D}$ | |
| LayerNorm (theory) | $\mu_n = \dfrac{1}{LD} \sum_{l=1}^{L} \sum_{d=1}^{D} \mathbf{X}_{n,l,d}$ $\sigma_n^2 = \dfrac{1}{LD} \sum_{l=1}^{L} \sum_{d=1}^{D} (\mathbf{X}_{n,l,d} - \mu_n)^2$ $\boldsymbol{\mu} = [\mu_1 \quad \mu_2 \quad \cdots \quad \mu_N]^\top \in \mathbb{R}^{N \times 1 \times 1}$ $\boldsymbol{\sigma}^2 = [\sigma_1^2 \quad \sigma_2^2 \quad \cdots \quad \sigma_N^2]^\top \in \mathbb{R}^{N \times 1 \times 1}$ | $\tilde{\mathbf{X}} = \dfrac{\mathbf{X} - \boldsymbol{\mu}}{\sqrt{\boldsymbol{\sigma}^2 + \varepsilon}}$ $\mathbf{Y} = \tilde{\mathbf{X}} \odot \boldsymbol{\gamma} + \boldsymbol{\beta}$ |
| LayerNorm (practice) | $\mu_{n,l} = \dfrac{1}{D} \sum_{d=1}^{D} \mathbf{X}_{n,l,d}$ $\sigma_{n,l}^2 = \dfrac{1}{D} \sum_{d=1}^{D} (\mathbf{X}_{n,l,d} - \mu_n)^2$ $\boldsymbol{\mu} = \begin{bmatrix} \mu_{1,1} & \mu_{1,2} & \cdots & \mu_{1,L} \\ \mu_{2,1} & \mu_{2,2} & \cdots & \mu_{2,L} \\ \vdots & \vdots & \ddots & \vdots \\ \mu_{N,1} & \mu_{N,2} & \cdots & \mu_{N,L} \end{bmatrix}^\top \in \mathbb{R}^{N \times L \times 1}$ $\boldsymbol{\sigma}^2 = \begin{bmatrix} \sigma_{1,1}^2 & \sigma_{1,2}^2 & \cdots & \sigma_{1,L}^2 \\ \sigma_{2,1}^2 & \sigma_{2,2}^2 & \cdots & \sigma_{2,L}^2 \\ \vdots & \vdots & \ddots & \vdots \\ \sigma_{N,1}^2 & \sigma_{N,2}^2 & \cdots & \sigma_{N,L}^2 \end{bmatrix}^\top \in \mathbb{R}^{N \times L \times 1}$ | |
| RMSNorm | $\|\mathbf{X}\|_{n,l} = \sqrt{\sum_{d=1}^{D} \mathbf{X}_{n,l,d}^2}$ | $\tilde{\mathbf{X}} = \sqrt{D} \dfrac{\mathbf{X}}{\|\mathbf{X}\|}$ $\mathbf{Y} = \tilde{\mathbf{X}} \odot \boldsymbol{\gamma} + \boldsymbol{\beta}$ |
| UnitNorm | $\|\mathbf{X}\| = \begin{bmatrix} \|\mathbf{X}\|_{1,1} & \|\mathbf{X}\|_{1,2} & \cdots & \|\mathbf{X}\|_{1,L} \\ \|\mathbf{X}\|_{2,1} & \|\mathbf{X}\|_{2,2} & \cdots & \|\mathbf{X}\|_{2,L} \\ \vdots & \vdots & \ddots & \vdots \\ \|\mathbf{X}\|_{N,1} & \|\mathbf{X}\|_{N,2} & \cdots & \|\mathbf{X}\|_{N,L} \end{bmatrix}^\top$ $\in \mathbb{R}^{N \times L \times 1}$ | $\tilde{\mathbf{X}} = D^{\frac{k}{2}} \dfrac{\mathbf{X}}{\|\mathbf{X}\|}$ $\mathbf{Y} = \tilde{\mathbf{X}}$ |

where $\mathbf{W}_Q, \mathbf{W}_K \in \mathbb{R}^{D \times D}$ are the projection matrices and $\mathbf{b}_Q, \mathbf{b}_K \in \mathbb{R}^D$ are the bias vectors for query and key, respectively. As the normalization and projection are both linear operations, we can combine them into a single linear operation as follows:

$$\begin{aligned}
\mathbf{Y} &= \tilde{\mathbf{X}} \mathbf{W}_Y + \mathbf{b}_Y \\
&= (\mathbf{XW} + \mathbf{b}) \, \mathbf{W}_Y + \mathbf{b}_Y \\
&= \mathbf{X} \, (\mathbf{W}\mathbf{W}_Y) + (\mathbf{b}\mathbf{W}_Y + \mathbf{b}_Y)
\end{aligned} \tag{33}$$

for $Y \in \{Q, K\}$. Therefore, there must exist some $\mathbf{W}'$, $\mathbf{b}'$, $\mathbf{W}'_Y$ and $\mathbf{b}'_Y$ such that:

$$\begin{aligned}
\mathbf{W}'_Y \mathbf{W}' &= \mathbf{W}\mathbf{W}_Y \\
\mathbf{b}_Y \mathbf{W}' + \mathbf{b}' &= \mathbf{b}\mathbf{W}_Y + \mathbf{b}_Y
\end{aligned} \tag{34}$$

Therefore, the order of normalization and projection does not affect the theoretical analysis. And in favor of simplicity, we can assume the normalization is performed after the projection.

## D  SUPPLEMENTARY FIGURES

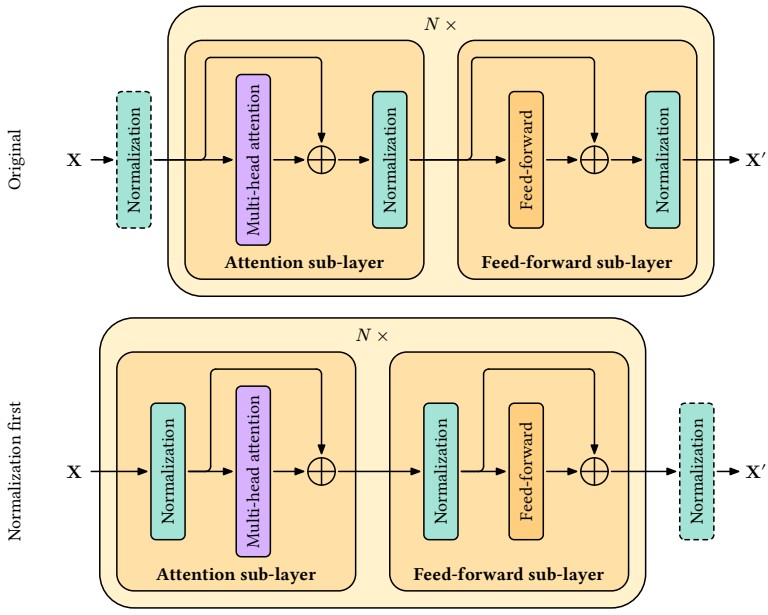

Figure S1: Transformer layer architecture. The original architecture is equivalent to a normalization-first sub-layer design for simpler analysis.

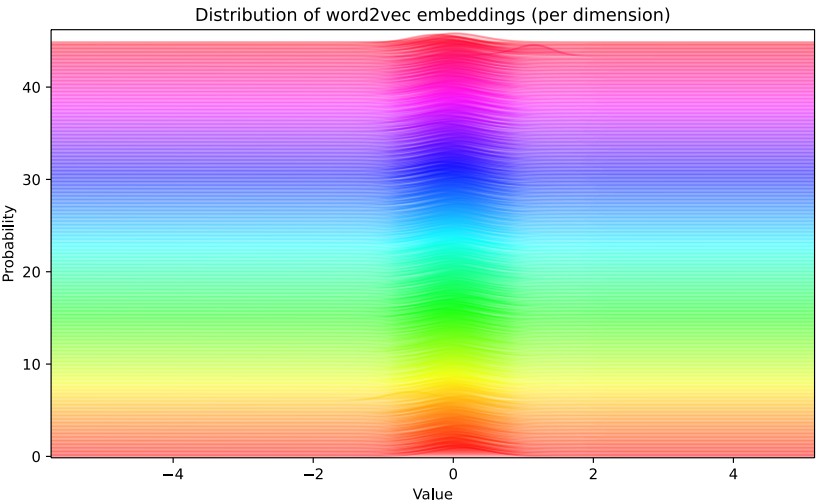

Figure S2: Distribution of values in each dimension of the word2vec embedding. The word2vec embedding is a 300-dimensional vector, and the distribution follows a normal distribution with means mostly around 0.

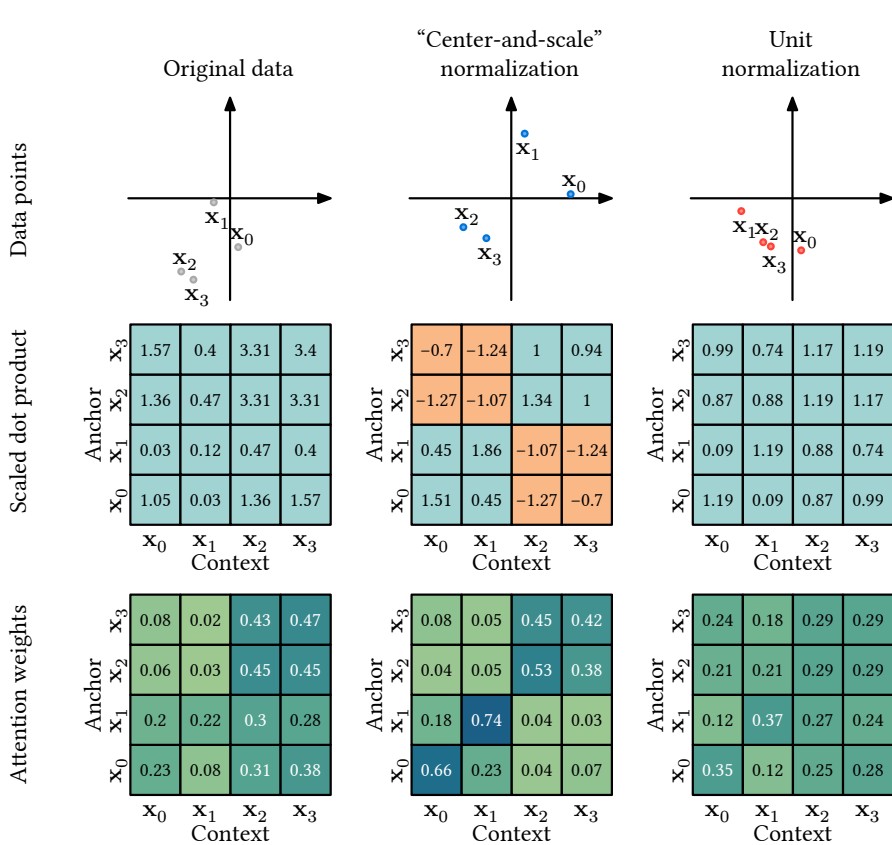

Figure S3: Demonstration of the token shift and attention shift problems using artificial data. The $\mathbf{x}_0$ and $\mathbf{x}_1$ exhibit typical token shift as shifting away from their original quadrants, resulting in sign flip in scaled dot product (marked in orange), and leading to less attention weights distributed to $\mathbf{x}_2$ and $\mathbf{x}_3$ than original. Attention shift and sparse attention problem can also be observed as the maximum attention weight is altered from $\mathbf{x}_2$ and $\mathbf{x}_3$ to nearly solely onto themselves.

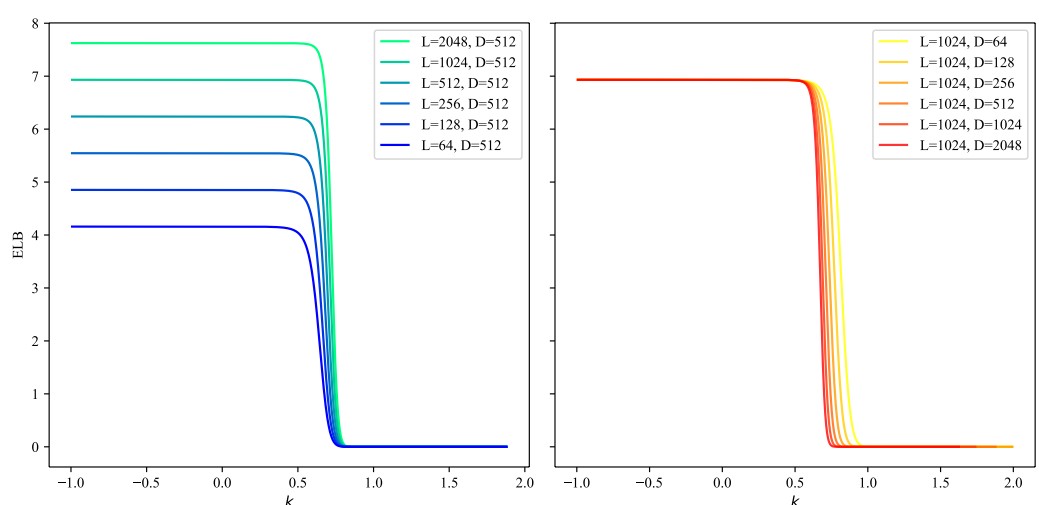

Figure S4: Entropy lower bound (ELB) against $k$ for different $L, D$. The left figure shows the curve for fixed $D = 512$ and varying $L$, and the right figure shows the curve for fixed $L = 1024$ and varying $D$.

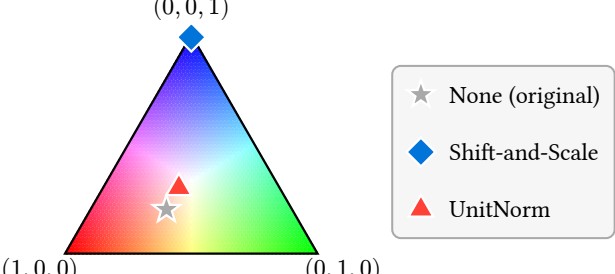

Figure S5: Graphical representation of attention weights showing a simple scenario of 3 tokens. Each corner represents a single-point distribution (red, blue and green) and the center representing a uniform distribution (white). Gray star, blue diamond, and red triangle mark the attention weights with no normalization, center-and-scale normalization, and UnitNorm, respectively.

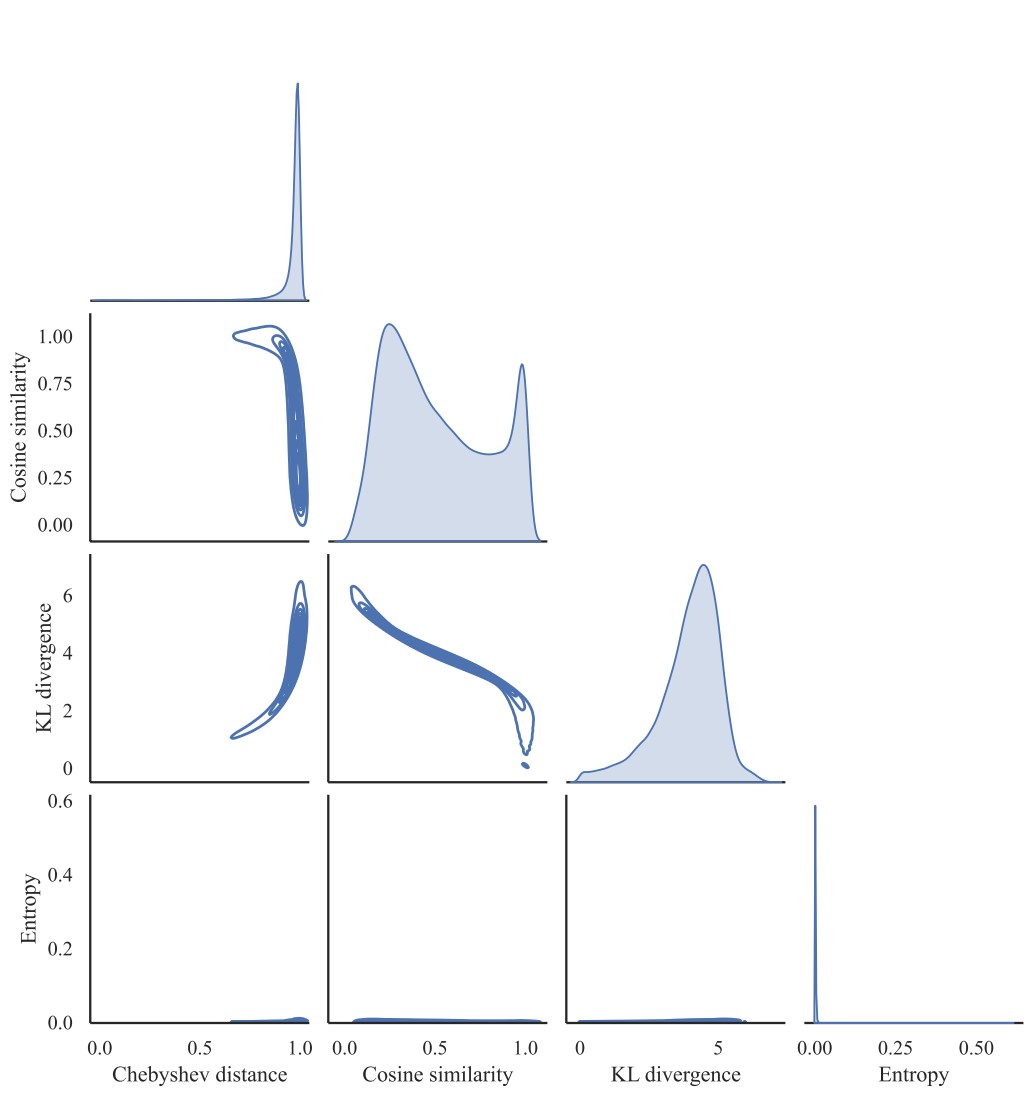

Figure S6: Joint distribution of metrics for LayerNorm (practice). Metrics used are defined as in Table S10.

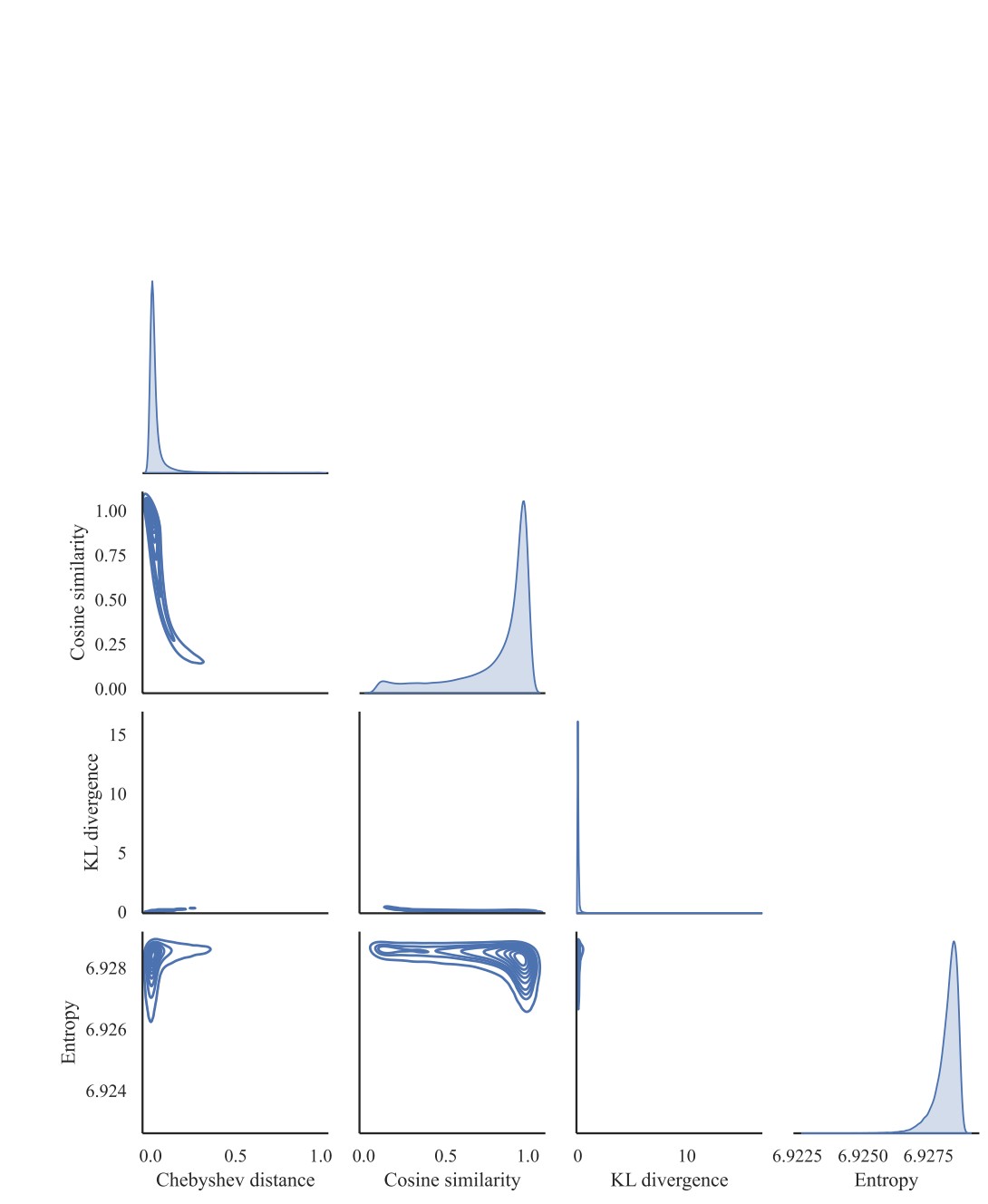

Figure S7: Joint distribution of metrics for UnitNorm. Metrics used are defined as in Table S10.

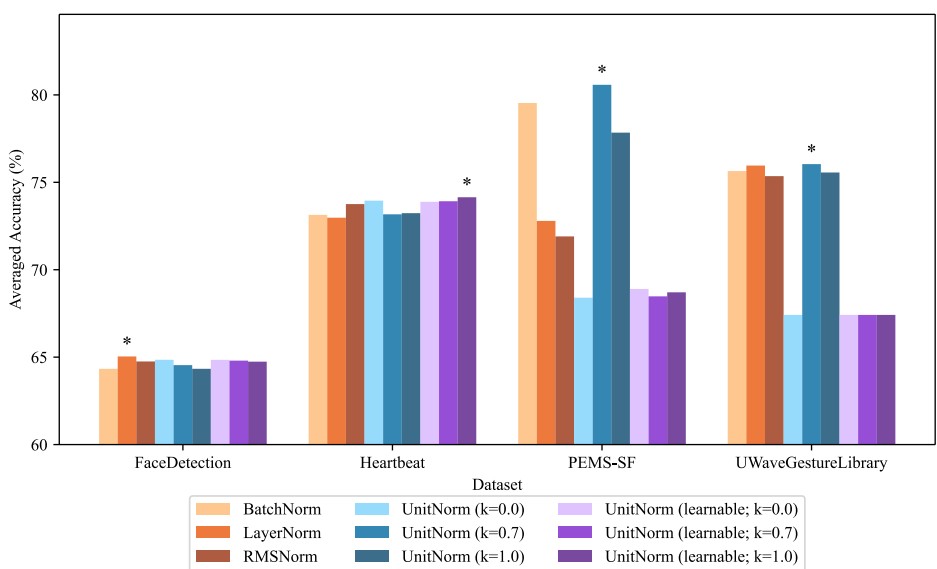

Figure S8: Average rank of normalization methods on the classification tasks. X-axis: Dataset with different normalization, Y-axis: average rank over models. Ranks are computed based on the accuracy of each model on each task with different normalization methods (lower is better). ∗ indicates the best performing normalization method(s) on each task. UnitNorm and UnitNorm (learnable) outperform other normalization methods on 3 out of 5 datasets, showing its potential in classification tasks.

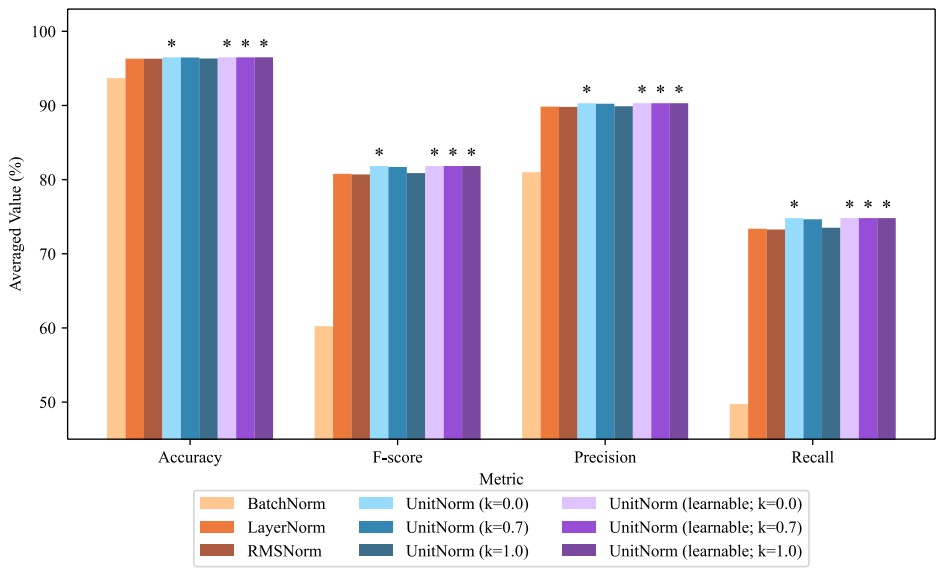

Figure S9: Average rank of normalization methods on the anomaly detection tasks. X-axis: Metrics under different normalization, Y-axis: average rank over models. Ranks are computed based on every metric of each model with different normalization methods (lower is better). ∗ indicates the best performing normalization method(s) on each metric. UnitNorm and UnitNorm (learnable) show a dominating performance gain over the other normalization methods in all metrics.

Table S2: Performance on a synthetic periodic dataset with two channels of sine waves (varying periods 2-5 Hz, amplitudes 0.5-2, plus Gaussian noise level = 0.1). Results shown for PatchTST model with different normalization methods, averaged over 3 random seeds with standard deviation. Best results are in **bold**.

| Normalization | MSE | MAE |
|---|---|---|
| BatchNorm | $2.721 \pm 3.409$ | $0.956 \pm 0.550$ |
| LayerNorm | $3.117 \pm 3.078$ | $1.054 \pm 0.532$ |
| RMSNorm | $3.114 \pm 3.057$ | $1.056 \pm 0.528$ |
| UnitNorm ($k = 0.5$) | $\mathbf{1.127 \pm 1.085}$ | $\mathbf{0.732 \pm 0.269}$ |
| UnitNorm ($k = 0.7$) | $1.600 \pm 1.762$ | $0.800 \pm 0.361$ |
| UnitNorm ($k = 1.0$) | $3.115 \pm 3.059$ | $1.056 \pm 0.528$ |

## E  SUPPLEMENTARY TABLES

Table S3: Summary of long term forecasting benchmark settings. The sequence length is the number of historical time steps fed into the encoder, and the label length is the number of time steps fed into the decoder as the ground truth output of the decoder. The prediction length is the number of time steps to be predicted by the decoder.

| Datasets | Feature number | Sequence length | Label length | Prediction length | Metrics | License |
|---|---|---|---|---|---|---|
| ETTh1, ETTh2 Zhou et al. (2021) | 7 | | | | | CC BY-ND 4.0 |
| ECL Trindade (2015) | 321 | 384 | 96 | {96, 192, 384, 720} | MSE, MAE | CC BY 4.0 |
| Exchange Lai et al. (2018) | 8 | | | | | N/A |

Table S4: Summary of classification benchmark settings. All datasets are from UEA Archive Bagnall et al. (2018). The sequence length is the number of time steps in each sequence fed into the encoder, and the prediction is made on the flattened output of the encoder by a fully connected layer.

| Datasets | Feature number | Class number | Sequence length | Metrics | License |
|---|---|---|---|---|---|
| FaceDetection | 144 | 5890 | | | |
| Heartbeat | 61 | 204 | 96 | Accuracy | N/A |
| PEMS-SF | 963 | 173 | | | |
| UWaveGestureLibrary | 3 | 320 | | | |

Table S5: Summary of anomaly detection benchmark settings. The sequence length is the number of time steps in each sequence fed into the model for reconstruction using MSE as loss. The threshold is determined by the distribution of reconstruction error on the training set, and the metrics are computed on the test set based on this threshold.

| Datasets | Feature number | Sequence length | Reconstruction error | Metrics | License |
|---|---|---|---|---|---|
| MSL Hundman et al. (2018) | 55 | 100 | MSE | Accuracy, F1-score, Precision, Recall | N/A |

Table S6: Summary of compute resources used for the experiments. Depending on the dataset and model, the GPU memory usage varies from 4G to 64G.

| CPU | Memory | GPU | GPU Memory |
|---|---|---|---|
| AMD Threadripper 3995WX | 512G | $4 \times$ NVIDIA RTX A5000 | $4 \times$ 24G |

Table S7: Long term forecasting test losses on different datasets using different models and normalization methods. For each dataset, prediction length, metric and for each model, the best performing normalization method(s) are bolded, and the second best are underlined.

| norm | model | ECL 96 MAE | 96 MSE | 192 MAE | 192 MSE | 336 MAE | 336 MSE | 720 MAE | 720 MSE | ETTh1 96 MAE | 96 MSE | 192 MAE | 192 MSE | 336 MAE | 336 MSE | 720 MAE | 720 MSE | ETTh2 96 MAE | 96 MSE | 192 MAE | 192 MSE | 336 MAE | 336 MSE | 720 MAE | 720 MSE | Exchange 96 MAE | 96 MSE | 192 MAE | 192 MSE | 336 MAE | 336 MSE | 720 MAE | 720 MSE |
|---|---|---|---|---|---|---|---|---|---|---|---|---|---|---|---|---|---|---|---|---|---|---|---|---|---|---|---|---|---|---|---|---|---|
| BatchNorm | Crossformer | 0.278 | 0.184 | 0.285 | 0.191 | 0.300 | 0.207 | 0.332 | 0.248 | **0.426** | 0.417 | 0.507 | 0.531 | 0.620 | 0.684 | 0.707 | 0.834 | **0.557** | **0.563** | 0.878 | 1.355 | 0.961 | 1.626 | 1.491 | 3.213 | **0.265** | **0.122** | **0.445** | **0.315** | **0.646** | **0.658** | **0.846** | **1.086** |
| BatchNorm | FEDformer | 0.435 | 0.357 | 0.448 | 0.364 | 0.455 | 0.376 | 0.478 | 0.417 | 0.422 | 0.389 | 0.449 | 0.433 | 0.477 | 0.483 | 0.507 | 0.516 | 0.434 | 0.423 | 0.473 | 0.469 | 0.483 | 0.486 | 0.507 | 0.479 | 0.297 | 0.169 | **0.383** | 0.276 | 0.487 | 0.438 | 0.823 | 1.151 |
| BatchNorm | Informer | 0.311 | 0.198 | 0.322 | 0.211 | 0.342 | 0.234 | 0.372 | 0.270 | 0.661 | 0.790 | 0.792 | 1.015 | 0.886 | 1.230 | 0.940 | 1.357 | 1.414 | 3.319 | 2.105 | 6.878 | 2.215 | 6.966 | 2.307 | 8.194 | 0.672 | 0.770 | 0.861 | 1.308 | 1.367 | 3.221 | 1.602 | 4.284 |
| BatchNorm | PatchTST | **0.246** | **0.144** | 0.265 | 0.162 | 0.290 | 0.189 | 0.338 | 0.255 | **0.395** | **0.374** | 0.429 | 0.426 | 0.462 | 0.478 | 0.497 | 0.506 | 0.346 | 0.292 | 0.399 | 0.376 | 0.435 | 0.421 | 0.453 | 0.435 | 0.204 | 0.088 | 0.300 | 0.180 | 0.419 | 0.336 | 0.703 | 0.874 |
| BatchNorm | Transformer | 0.408 | 0.310 | 0.409 | 0.321 | 0.463 | 0.407 | 0.510 | 0.461 | 0.814 | 1.008 | 0.805 | 1.014 | 0.886 | 1.161 | 0.819 | 1.054 | 1.093 | 1.852 | 1.258 | 2.488 | 1.603 | 3.874 | 1.833 | 4.782 | 0.608 | 0.665 | 0.921 | 1.458 | 1.242 | 2.578 | 1.444 | 3.069 |
| LayerNorm | Crossformer | **0.270** | **0.181** | **0.309** | **0.195** | 0.324 | 0.211 | 0.347 | 0.249 | **0.420** | 0.439 | **0.379** | 0.419 | **0.444** | 0.500 | **0.420** | 0.511 | **0.447** | **0.572** | **0.460** | **0.621** | 0.492 | 0.667 | 0.511 | 0.765 | 0.297 | 0.378 | 0.169 | 0.274 | 0.488 | 0.890 | 0.824 | 1.153 |
| LayerNorm | FEDformer | **0.276** | **0.187** | 0.324 | 0.211 | 0.347 | 0.236 | 0.366 | 0.262 | **0.420** | 0.395 | **0.444** | 0.351 | 0.471 | 0.438 | 0.464 | 0.507 | 0.423 | 0.425 | 0.475 | 0.464 | **0.469** | 0.488 | 0.481 | 0.507 | 0.209 | 0.091 | 0.383 | 0.169 | 0.488 | 0.440 | 0.824 | 1.153 |
| LayerNorm | Informer | 0.499 | 0.291 | 0.443 | 0.364 | 0.455 | 0.325 | 0.391 | 0.455 | 0.720 | 0.877 | 0.792 | 1.013 | 0.842 | 1.148 | **0.855** | 1.174 | 1.502 | 3.574 | 2.056 | 6.156 | 1.864 | 4.990 | 1.652 | 3.943 | 0.733 | 0.830 | 0.836 | 1.081 | 1.073 | 1.817 | 1.409 | 2.967 |
| LayerNorm | PatchTST | 0.358 | 0.187 | 0.355 | 0.211 | 0.347 | 0.204 | 0.391 | 0.246 | 0.396 | 0.377 | 0.426 | 0.422 | **0.447** | 0.460 | 0.492 | 0.511 | 0.345 | 0.297 | 0.399 | 0.380 | 0.431 | 0.417 | 0.447 | 0.429 | 0.209 | 0.091 | 0.301 | 0.180 | 0.412 | 0.324 | 0.713 | 0.901 |
| LayerNorm | Transformer | 0.542 | 0.325 | 0.535 | 0.366 | 0.455 | 0.376 | 0.514 | 0.391 | 0.884 | 1.188 | 0.802 | 1.820 | 0.834 | 5.161 | 0.833 | 1.078 | 2.205 | 5.161 | 1.820 | 6.156 | 1.699 | 4.549 | 1.390 | 2.908 | 0.589 | 0.581 | 0.786 | 1.047 | 1.051 | 1.745 | 1.212 | 2.297 |
| RMSNorm | Crossformer | 0.403 | 0.249 | 0.417 | 0.334 | 0.444 | 0.366 | 0.490 | 0.448 | 0.421 | 0.431 | 0.380 | 0.409 | 0.445 | 0.495 | 0.420 | 0.508 | 0.709 | 0.709 | 0.864 | 1.112 | 0.822 | 0.857 | 1.172 | 0.836 | 0.297 | 0.371 | 0.169 | 0.261 | 0.383 | 0.522 | 0.824 | 1.695 |
| RMSNorm | FEDformer | 0.303 | 0.150 | 0.324 | 0.262 | 0.346 | 0.211 | 0.365 | 0.261 | 0.393 | 0.350 | 0.445 | 0.420 | 0.471 | 0.462 | 0.507 | 0.505 | 0.393 | 0.350 | 0.437 | 0.425 | 0.469 | 0.488 | 0.481 | 0.481 | 0.209 | 0.091 | 0.383 | 0.277 | 0.488 | 0.440 | 0.824 | 1.153 |
| RMSNorm | Informer | 0.427 | 0.292 | 0.444 | 0.211 | 0.460 | 0.365 | 0.399 | 0.264 | 0.709 | 0.864 | 0.781 | 0.995 | 0.822 | 1.112 | 0.857 | 1.172 | 1.573 | 4.058 | 2.126 | 6.495 | 1.941 | 5.358 | 1.663 | 3.887 | 0.730 | 0.841 | 0.866 | 1.158 | 1.077 | 1.840 | 1.386 | 2.875 |
| RMSNorm | PatchTST | 0.343 | 0.182 | 0.366 | 0.211 | 0.346 | 0.235 | 0.327 | 0.247 | 0.377 | 0.425 | 0.421 | 0.448 | 0.463 | 0.495 | 0.505 | 0.520 | 0.345 | 0.297 | 0.399 | 0.379 | 0.431 | 0.418 | 0.448 | 0.429 | 0.209 | 0.091 | 0.300 | 0.179 | 0.411 | 0.322 | 0.715 | 0.904 |
| RMSNorm | Transformer | 0.490 | 0.204 | 0.448 | 0.370 | 0.539 | 0.460 | 0.511 | 0.399 | 0.745 | 0.882 | 0.802 | 0.984 | 0.870 | 1.123 | 0.808 | 1.030 | 1.211 | 2.276 | 1.815 | 5.030 | 1.806 | 5.043 | 1.480 | 3.239 | 0.580 | 0.569 | 0.783 | 1.046 | 1.045 | 1.742 | 1.246 | 2.354 |
| UnitNorm (k=0.0) | Crossformer | 0.445 | 0.282 | 0.499 | 0.295 | 0.530 | 0.309 | 0.535 | 0.348 | 0.591 | 0.686 | 0.607 | 0.705 | 0.607 | 0.709 | 0.649 | 0.759 | 0.576 | 0.694 | **0.756** | **1.214** | **0.811** | **1.297** | 1.071 | 1.982 | 0.688 | 0.913 | 0.772 | 1.057 | 1.177 | 2.187 | 1.338 | 2.711 |
| UnitNorm (k=0.0) | FEDformer | 0.280 | 0.184 | 0.325 | 0.211 | 0.368 | 0.212 | 0.398 | 0.257 | 0.440 | 0.410 | 0.457 | 0.440 | 0.477 | 0.472 | 0.503 | 0.497 | 0.399 | 0.355 | 0.441 | 0.430 | 0.480 | 0.476 | 0.490 | 0.486 | 0.298 | 0.170 | 0.385 | 0.279 | 0.487 | 0.438 | 0.820 | 1.145 |
| UnitNorm (k=0.0) | Informer | 0.499 | 0.345 | 0.520 | 0.265 | 0.535 | 0.517 | 0.517 | 0.440 | 0.816 | 1.104 | 0.814 | 1.111 | 0.843 | 1.141 | 0.891 | 1.212 | 1.155 | 2.060 | 1.258 | 2.582 | 1.287 | 2.396 | 1.306 | 2.447 | 1.016 | 1.497 | 1.066 | 1.646 | 1.186 | 2.042 | 1.200 | 2.155 |
| UnitNorm (k=0.0) | PatchTST | 0.457 | 0.193 | 0.485 | 0.238 | 0.497 | 0.265 | 0.517 | 0.303 | 0.401 | 0.384 | 0.431 | 0.431 | 0.453 | 0.471 | 0.483 | 0.503 | 0.341 | 0.290 | 0.394 | 0.369 | **0.429** | 0.417 | 0.449 | 0.429 | 0.206 | 0.088 | **0.299** | **0.178** | 0.418 | 0.331 | 0.707 | 0.876 |
| UnitNorm (k=0.0) | Transformer | 0.505 | 0.454 | 0.498 | 0.435 | 0.800 | 0.990 | 0.791 | 0.979 | 0.842 | 1.076 | 0.844 | 1.077 | 0.999 | 1.518 | 1.189 | 2.231 | 1.248 | 2.358 | 1.291 | 2.379 | 1.023 | 1.535 | 1.121 | 1.848 | 1.203 | 2.122 | 1.111 | 1.717 | 0.483 | 0.486 | 0.341 | 0.290 |

| dataset | ECL | | | | | | | | ETTh1 | | | | | | | | ETTh2 | | | | | | | | Exchange | | | | | | | |
|---|---|---|---|---|---|---|---|---|---|---|---|---|---|---|---|---|---|---|---|---|---|---|---|---|---|---|---|---|---|---|---|---|
| prediction length | 96 | | 192 | | 336 | | 720 | | 96 | | 192 | | 336 | | 720 | | 96 | | 192 | | 336 | | 720 | | 96 | | 192 | | 336 | | 720 | |
| metric | MAE | MSE | MAE | MSE | MAE | MSE | MAE | MSE | MAE | MSE | MAE | MSE | MAE | MSE | MAE | MSE | MAE | MSE | MAE | MSE | MAE | MSE | MAE | MSE | MAE | MSE | MAE | MSE | MAE | MSE | MAE | MSE |
| **UnitNorm (k=0.5)** Crossformer | 0.420 | 0.257 | 0.463 | 0.274 | 0.492 | 0.299 | 0.501 | **0.339** | 0.436 | 0.408 | 0.498 | 0.499 | 0.600 | 0.643 | 0.673 | 0.759 | 0.657 | 0.863 | 0.832 | 1.326 | 1.120 | 2.004 | 1.369 | 2.851 | 0.472 | 0.430 | 0.644 | 0.702 | 0.964 | 1.526 | 1.128 | 1.971 |
| FEDformer | 0.331 | 0.204 | 0.335 | 0.222 | 0.354 | 0.242 | 0.385 | 0.283 | 0.428 | 0.386 | 0.448 | 0.423 | 0.471 | 0.462 | 0.502 | 0.497 | 0.401 | 0.356 | 0.438 | 0.427 | 0.480 | 0.475 | **0.483** | **0.474** | 0.298 | 0.170 | 0.385 | 0.279 | 0.487 | 0.438 | 0.820 | 1.145 |
| Informer | 0.446 | 0.280 | 0.492 | 0.434 | 0.501 | 0.441 | 0.558 | 0.543 | 0.748 | 0.963 | 0.751 | 0.981 | 0.800 | 1.061 | 0.871 | 1.190 | 1.099 | 0.401 | 1.884 | 0.356 | 3.014 | 1.343 | 2.696 | 2.927 | 0.838 | 1.068 | 0.983 | 1.457 | 1.028 | 1.595 | 1.023 | 1.608 |
| PatchTST | 0.371 | 0.189 | 0.296 | 0.205 | 0.328 | 0.247 | 0.397 | 0.381 | 0.662 | 0.751 | 0.736 | 0.917 | 0.821 | 1.073 | 0.874 | 1.186 | 0.343 | 0.291 | 0.393 | 0.365 | 0.433 | 0.480 | 0.449 | 0.431 | 0.202 | 0.085 | 0.300 | 0.178 | 0.411 | 0.321 | 0.698 | 0.861 |
| Transformer | 0.504 | 0.300 | 0.460 | 0.209 | 0.508 | 0.332 | 0.459 | 0.251 | 0.914 | 1.243 | 1.391 | 3.050 | 1.356 | 2.854 | 2.669 | 0.782 | 0.914 | 0.657 | 1.243 | 0.863 | 1.391 | 1.326 | 2.854 | 2.004 | 0.965 | 0.915 | 1.328 | 0.993 | 1.484 | 0.911 | 1.196 | 1.971 |
| **UnitNorm (k=1.0)** Crossformer | _0.405_ | _0.271_ | _0.309_ | 0.182 | _0.277_ | 0.188 | 0.292 | 0.204 | 0.396 | 0.377 | **0.425** | 0.421 | 0.448 | 0.463 | 0.495 | 0.519 | 0.345 | 0.297 | 0.399 | 0.380 | 0.430 | 0.469 | 0.446 | 0.426 | 0.209 | 0.091 | 0.300 | 0.179 | 0.411 | 0.322 | 0.715 | 0.904 |
| FEDformer | _0.304_ | 0.182 | _0.195_ | 0.211 | _0.249_ | 0.165 | **0.290** | 0.190 | 0.421 | 0.380 | 0.445 | 0.420 | 0.471 | 0.462 | 0.501 | 0.496 | 0.393 | 0.350 | 0.437 | 0.425 | 0.475 | 0.488 | 0.481 | 0.481 | 0.297 | 0.169 | 0.383 | 0.277 | 0.488 | 0.440 | 0.824 | 1.153 |
| Informer | 0.428 | 0.277 | 0.324 | 0.346 | 0.346 | 0.235 | 0.365 | 0.341 | 0.713 | 0.869 | 0.780 | 0.992 | 0.823 | 1.113 | 0.849 | 1.160 | 1.559 | 3.951 | 2.119 | 6.448 | 1.849 | 4.921 | 3.883 | 2.491 | 0.877 | 0.860 | 1.137 | 1.841 | 1.396 | 2.936 | 2.354 | 2.659 |
| PatchTST | 0.292 | 0.204 | 0.368 | 0.235 | **0.346** | 0.261 | 0.365 | 0.247 | 0.396 | 0.377 | 0.421 | 0.420 | 0.448 | 0.462 | 0.463 | 0.495 | 0.341 | 0.291 | 0.399 | 0.380 | 0.430 | 0.469 | 0.446 | 0.426 | 0.202 | 0.091 | 0.300 | 0.179 | 0.411 | 0.322 | 0.715 | 0.904 |
| Transformer | 0.511 | 0.247 | 0.346 | 0.368 | 0.442 | 0.368 | 0.540 | 0.464 | 0.745 | 0.881 | 0.802 | 0.984 | 0.871 | 1.125 | 0.806 | 1.026 | 1.213 | 2.283 | 1.818 | 5.050 | 1.802 | 5.019 | 1.480 | 3.242 | 0.569 | 0.783 | 1.046 | 1.046 | 1.742 | 1.246 | 2.354 | 1.702 |
| **UnitNorm (learnable; k=0.0)** Crossformer | 0.449 | 0.280 | 0.502 | 0.324 | 0.282 | 0.461 | 0.211 | 0.344 | 0.780 | 0.400 | 0.963 | 0.384 | 0.808 | 0.430 | 1.010 | 0.431 | 0.964 | 0.399 | 1.393 | 0.291 | 1.216 | 0.394 | 2.312 | 0.370 | 0.982 | 0.204 | 1.426 | 0.086 | 1.097 | 0.300 | 1.772 | 0.179 |
| FEDformer | 0.361 | 0.189 | 0.461 | 0.211 | 0.184 | 0.508 | 0.237 | 0.367 | 0.963 | 0.410 | 0.820 | 0.457 | 1.120 | 0.439 | 0.844 | 0.477 | 1.059 | 1.722 | 1.237 | 2.430 | 1.259 | 2.273 | 1.327 | 2.491 | 0.986 | 1.438 | 1.029 | 1.541 | 1.174 | 2.001 | 1.147 | 1.963 |
| Informer | 0.459 | 0.284 | 0.508 | 0.293 | 0.298 | 0.539 | 0.265 | 0.398 | 0.850 | 0.440 | 1.169 | 0.410 | 0.820 | 0.477 | 1.149 | 0.470 | 0.341 | 0.291 | 0.441 | 0.430 | 0.428 | 0.416 | 0.448 | 0.428 | 0.298 | 0.170 | 0.385 | 0.279 | 0.487 | 0.438 | 0.820 | 1.145 |
| PatchTST | 0.388 | 0.193 | 0.460 | 0.237 | 0.198 | 0.526 | 0.208 | 0.304 | 1.010 | 0.430 | 0.845 | 0.439 | 0.849 | 0.453 | 1.101 | 0.471 | 0.394 | 0.430 | 1.237 | 1.324 | 1.316 | 1.058 | 1.940 | 0.663 | 0.298 | 0.385 | 0.487 | 0.438 | 0.820 | 1.145 | 2.659 | 0.841 |
| Transformer | 0.436 | 0.492 | 0.300 | 0.209 | 0.493 | 0.331 | 0.427 | 0.251 | 0.817 | 0.483 | 0.894 | 0.508 | 0.486 | 0.503 | 1.036 | 1.228 | 1.228 | 2.284 | 1.416 | 2.273 | 1.306 | 1.490 | 2.386 | 0.428 | 1.192 | 2.084 | 1.079 | 1.618 | 0.882 | 0.882 | 1.145 | 2.659 |
| **UnitNorm (learnable; k=0.7)** Crossformer | 0.445 | 0.280 | 0.500 | 0.325 | 0.282 | 0.461 | 0.211 | 0.345 | 0.813 | 0.401 | 1.003 | 0.384 | 0.828 | 0.431 | 1.037 | 0.431 | 0.985 | 0.341 | 1.495 | 0.290 | 1.160 | 0.394 | 2.112 | 0.369 | 1.023 | 0.202 | 1.535 | 0.084 | 1.121 | 0.300 | 1.848 | 0.180 |
| FEDformer | 0.359 | 0.189 | 0.461 | 0.211 | 0.184 | 0.520 | 0.238 | 0.368 | 1.107 | 0.410 | 0.807 | 0.456 | 1.094 | 0.437 | 0.845 | 0.480 | 1.093 | 1.861 | 1.251 | 2.407 | 1.284 | 2.388 | 1.306 | 2.427 | 1.031 | 1.557 | 1.075 | 1.705 | 1.186 | 2.042 | 1.163 | 2.083 |
| Informer | 0.460 | 0.284 | 0.520 | 0.295 | 0.200 | 0.530 | 0.265 | 0.394 | 0.815 | 0.456 | 0.807 | 0.437 | 0.845 | 0.476 | 1.141 | 0.483 | 0.341 | 0.290 | 0.445 | 0.434 | 0.475 | 0.485 | 0.480 | 0.429 | 0.293 | 0.164 | 0.385 | 0.280 | 0.487 | 0.438 | 0.830 | 1.167 |
| PatchTST | 0.392 | 0.193 | 0.486 | 0.238 | 0.309 | 0.497 | 0.212 | 0.300 | 1.094 | 0.437 | 0.845 | 0.453 | 1.141 | 0.471 | 0.887 | 0.483 | 0.290 | 0.369 | 2.407 | 1.284 | 1.379 | 1.041 | 1.914 | 0.688 | 0.913 | 0.772 | 1.057 | 1.177 | 2.187 | 1.338 | 2.711 | 0.797 |
| Transformer | 0.505 | 0.454 | 0.300 | 0.209 | 0.498 | 0.332 | 0.435 | 0.252 | 0.854 | 0.483 | 0.888 | 0.509 | 0.486 | 0.504 | 1.100 | 1.209 | 1.229 | 2.112 | 1.229 | 2.378 | 1.290 | 1.306 | 2.399 | 0.428 | 1.203 | 2.122 | 1.111 | 1.717 | 0.869 | 0.869 | 1.167 | 2.711 |
| **UnitNorm (learnable; k=1.0)** Crossformer | 0.445 | 0.280 | 0.502 | 0.325 | 0.282 | 0.462 | 0.211 | 0.345 | 0.813 | 0.401 | 1.003 | 0.384 | 0.828 | 0.431 | 1.037 | 0.431 | 0.985 | 0.341 | 1.495 | 0.290 | 1.160 | 0.394 | 2.112 | 0.369 | 1.023 | 0.202 | 1.535 | 0.084 | 1.121 | 0.300 | 1.848 | 0.180 |
| FEDformer | 0.359 | 0.189 | 0.462 | 0.211 | 0.184 | 0.522 | 0.238 | 0.368 | 1.104 | 0.410 | 0.810 | 0.456 | 1.112 | 0.437 | 0.846 | 0.480 | 1.119 | 1.917 | 1.233 | 2.415 | 1.268 | 2.314 | 1.305 | 2.436 | 1.021 | 1.527 | 1.081 | 1.725 | 1.186 | 2.042 | 1.150 | 2.031 |
| Informer | 0.460 | 0.522 | 0.295 | 0.490 | 0.531 | 0.309 | 0.547 | 0.265 | 0.817 | 0.817 | 0.810 | 0.437 | 0.846 | 0.476 | 1.142 | 0.483 | 0.341 | 0.290 | 0.445 | 0.434 | 0.480 | 0.485 | 0.480 | 0.429 | 0.293 | 0.164 | 0.385 | 0.280 | 0.487 | 0.438 | 0.830 | 1.167 |
| PatchTST | 0.392 | 0.490 | 0.368 | 0.499 | 0.547 | 0.394 | 0.429 | 0.300 | 1.112 | 0.437 | 0.846 | 0.453 | 1.142 | 0.471 | 0.888 | 0.483 | 0.290 | 0.369 | 1.233 | 1.268 | 1.379 | 1.041 | 1.914 | 0.688 | 0.202 | 0.084 | 0.300 | 0.180 | 0.418 | 0.331 | 0.703 | 0.869 |
| Transformer | 0.505 | 0.499 | 0.300 | 0.209 | 0.498 | 0.331 | 0.435 | 0.252 | 0.854 | 0.483 | 0.888 | 0.509 | 0.486 | 0.504 | 1.100 | 1.211 | 1.119 | 2.112 | 1.229 | 2.378 | 1.290 | 1.305 | 2.399 | 2.436 | 1.081 | 1.725 | 1.186 | 2.042 | 1.150 | 2.031 | 0.869 | 2.711 |

Table S8: Classification accuracies of different datasets using different models and normalization methods. For each dataset and for each model, the best performing normalization method(s) are bolded, and the second best are underlined.

| | dataset | Face Detection | Heartbeat | PEMS-SF | UWave Gesture Library |
|---|---|---|---|---|---|
| **BatchNorm** | Crossformer | 50.435 | **75.122** | **68.401** | **83.438** |
| | FEDformer | 68.275 | 73.984 | 78.035 | 47.812 |
| | Informer | 68.606 | 73.984 | **87.476** | 82.083 |
| | PatchTST | 65.683 | 66.992 | 79.576 | 81.354 |
| | Transformer | 68.663 | 75.610 | 84.200 | 83.542 |
| **LayerNorm** | Crossformer | **52.176** | 73.008 | 26.397 | 82.708 |
| | FEDformer | 68.861 | 73.659 | 84.393 | 48.438 |
| | Informer | 68.076 | 73.821 | 85.742 | 81.979 |
| | PatchTST | 67.329 | 69.919 | **84.971** | 81.562 |
| | Transformer | 68.757 | 74.472 | 82.466 | 85.104 |
| **RMSNorm** | Crossformer | 51.693 | 73.659 | 23.699 | 81.042 |
| | FEDformer | 68.000 | 72.846 | 85.164 | 47.708 |
| | Informer | 68.275 | 75.447 | 84.586 | 83.125 |
| | PatchTST | 66.648 | 70.894 | 82.852 | 80.729 |
| | Transformer | **69.154** | **75.935** | 83.237 | 84.167 |
| **UnitNorm (k=0.0)** | Crossformer | 50.000 | 72.195 | 16.763 | 29.167 |
| | FEDformer | **69.041** | **74.146** | 83.430 | **53.333** |
| | Informer | **69.088** | 75.285 | 78.035 | **85.000** |
| | PatchTST | 67.546 | 72.195 | 81.118 | 82.396 |
| | Transformer | 68.568 | **75.935** | 82.659 | **87.188** |
| **UnitNorm (k=0.7)** | Crossformer | 50.236 | 73.008 | 65.896 | 81.667 |
| | FEDformer | 68.067 | 72.846 | 84.586 | 47.083 |
| | Informer | 68.142 | 72.846 | 83.237 | 83.958 |
| | PatchTST | **67.641** | 72.358 | **84.971** | 82.604 |
| | Transformer | 68.634 | 74.797 | **84.200** | 84.896 |
| **UnitNorm (k=1.0)** | Crossformer | 50.019 | 72.195 | 56.455 | 80.521 |
| | FEDformer | 67.357 | 73.984 | **85.356** | 48.125 |
| | Informer | 68.492 | 73.984 | 84.008 | 83.646 |
| | PatchTST | 67.452 | 72.033 | 83.044 | 81.146 |
| | Transformer | 68.350 | 73.984 | 80.347 | 84.375 |
| **UnitNorm (learnable; k=0.0)** | Crossformer | 50.000 | 72.195 | 16.763 | 29.167 |
| | FEDformer | **69.041** | **74.146** | 83.430 | **53.333** |
| | Informer | 69.079 | 74.959 | 80.539 | **85.000** |
| | PatchTST | 67.546 | 72.195 | 81.118 | 82.396 |
| | Transformer | 68.568 | **75.935** | 82.659 | **87.188** |
| **UnitNorm (learnable; k=0.7)** | Crossformer | 50.000 | 72.195 | 16.763 | 29.167 |
| | FEDformer | **69.041** | **74.146** | 83.430 | **53.333** |
| | Informer | 68.852 | 75.122 | 78.420 | **85.000** |
| | PatchTST | 67.546 | 72.195 | 81.118 | 82.396 |
| | Transformer | 68.568 | **75.935** | 82.659 | **87.188** |
| **UnitNorm (learnable; k=1.0)** | Crossformer | 50.000 | 72.195 | 16.763 | 29.167 |
| | FEDformer | **69.041** | **74.146** | 83.430 | **53.333** |
| | Informer | 68.558 | **75.610** | 79.576 | **85.000** |
| | PatchTST | 67.546 | **72.846** | 81.118 | 82.396 |
| | Transformer | 68.568 | **75.935** | 82.659 | **87.188** |

Table S9: Anomaly detection accuracies of MSL dataset using different models and normalization methods. For each metric and for each model, the best performing normalization method(s) are bolded, and the second best are underlined.

| | metric | Accuracy | F-score | Precision | Recall |
|---|---|---|---|---|---|
| **BatchNorm** | Crossformer | 93.507 | 59.110 | 81.410 | 47.903 |
| | FEDformer | 95.543 | 75.820 | 88.630 | 66.240 |
| | Informer | 93.040 | 56.927 | 81.760 | 43.733 |
| | PatchTST | 95.947 | 78.613 | 88.603 | 70.650 |
| | Transformer | 90.417 | 30.680 | 64.623 | 20.123 |
| **LayerNorm** | Crossformer | 96.313 | 80.640 | 90.330 | 72.823 |
| | FEDformer | **96.603** | **82.427** | **90.697** | **75.537** |
| | Informer | 96.390 | 81.193 | 90.120 | 73.877 |
| | PatchTST | 95.950 | 78.727 | 88.347 | 70.993 |
| | Transformer | 96.333 | 80.910 | 89.740 | 73.660 |
| **RMSNorm** | Crossformer | 96.307 | 80.613 | 90.323 | 72.790 |
| | FEDformer | 96.573 | 82.263 | 90.647 | 75.300 |
| | Informer | 96.373 | 81.067 | 90.097 | 73.680 |
| | PatchTST | 95.940 | 78.670 | 88.307 | 70.927 |
| | Transformer | 96.330 | 80.883 | 89.677 | 73.657 |
| **UnitNorm (k=0.0)** | Crossformer | **96.533** | **82.000** | 90.610 | **74.880** |
| | FEDformer | 96.547 | 82.097 | 90.653 | 75.013 |
| | Informer | 96.543 | 82.067 | 90.640 | 74.973 |
| | PatchTST | **96.317** | **80.943** | **88.990** | **74.227** |
| | Transformer | 96.540 | 82.060 | 90.637 | **74.960** |
| **UnitNorm (k=0.7)** | Crossformer | 96.523 | 81.943 | **90.617** | 74.783 |
| | FEDformer | 96.537 | 82.040 | 90.620 | 74.943 |
| | Informer | **96.557** | **82.147** | 90.643 | **75.103** |
| | PatchTST | 96.213 | 80.360 | 88.713 | 73.440 |
| | Transformer | **96.540** | **82.043** | 90.580 | 74.977 |
| **UnitNorm (k=1.0)** | Crossformer | 96.347 | 80.857 | 90.380 | 73.143 |
| | FEDformer | 96.587 | 82.350 | 90.670 | 75.433 |
| | Informer | 96.470 | 81.653 | 90.400 | 74.450 |
| | PatchTST | 95.933 | 78.607 | 88.273 | 70.847 |
| | Transformer | 96.337 | 80.933 | 89.753 | 73.687 |
| **UnitNorm (learnable; k=0.0)** | Crossformer | **96.533** | 82.000 | 90.610 | **74.880** |
| | FEDformer | 96.547 | 82.097 | 90.653 | 75.013 |
| | Informer | 96.543 | 82.067 | 90.640 | 74.973 |
| | PatchTST | **96.317** | 80.943 | **88.990** | 74.227 |
| | Transformer | 96.540 | 82.060 | 90.637 | 74.960 |
| **UnitNorm (learnable; k=0.7)** | Crossformer | **96.533** | **82.000** | 90.610 | **74.880** |
| | FEDformer | 96.547 | 82.097 | 90.653 | 75.013 |
| | Informer | 96.543 | 82.067 | 90.640 | 74.973 |
| | PatchTST | **96.317** | 80.943 | **88.990** | **74.227** |
| | Transformer | 96.540 | 82.060 | 90.637 | 74.960 |
| **UnitNorm (learnable; k=1.0)** | Crossformer | **96.533** | **82.000** | 90.610 | **74.880** |
| | FEDformer | 96.547 | 82.097 | 90.653 | 75.013 |
| | Informer | 96.543 | 82.067 | 90.640 | 74.973 |
| | PatchTST | **96.317** | **80.943** | **88.990** | **74.227** |
| | Transformer | 96.540 | **82.060** | 90.637 | 74.960 |

Table S10: Metrics used for characterizing the distribution of attention scores from the original and normalized data, denoted as $\mathbf{A}_{n,i}$ and $\tilde{\mathbf{A}}_{n,i}$, respectively, for the $i$-th sample in the $n$-th batch.

| Metric | Definition | Evaluation |
|---|---|---|
| Chebyshev distance | $D_{\text{Chebyshev}}\left(\mathbf{A}_{n,i}, \tilde{\mathbf{A}}_{n,i}\right) = \max_{j=1}^{L}\left|\mathbf{A}_{n,i,j} - \tilde{\mathbf{A}}_{n,i,j}\right|$ | Lower is Better |
| Cosine similarity | $D_{\text{Cosine}}\left(\mathbf{A}_{n,i}, \tilde{\mathbf{A}}_{n,i}\right) = \frac{\mathbf{A}_{n,i}^{\top}\tilde{\mathbf{A}}_{n,i}}{\|\mathbf{A}_{n,i}\|\|\tilde{\mathbf{A}}_{n,i}\|}$ | Higher is Better |
| KL divergence | $D_{\text{KL}}\left(\mathbf{A}_{n,i}\|\tilde{\mathbf{A}}_{n,i}\right) = \sum_{j=1}^{L}\mathbf{A}_{n,i,j}\left(\log\mathbf{A}_{n,i,j} - \log\tilde{\mathbf{A}}_{n,i,j}\right)$ | Lower is Better |
| Entropy | $E\left(\tilde{\mathbf{A}}_{n,i}\right) = -\sum_{j=1}^{L}\tilde{\mathbf{A}}_{n,i,j}\log\tilde{\mathbf{A}}_{n,i,j}$ | Higher is Better |

Table T1: Last-value predictor versus PatchTST with different normalization methods on Exchange dataset. Best results *aside from last-value predictor* are in **bold**.

| Model & Metric\Prediction | 96 | 192 | 336 | 720 |
|---|---|---|---|---|
| Last-value predictor (MAE) | 0.196 | 0.289 | 0.398 | 0.676 |
| PatchTST + UnitNorm (MAE) | **0.202** | **0.300** | **0.411** | **0.698** |
| PatchTST + BatchNorm (MAE) | 0.204 | 0.300 | 0.419 | 0.703 |
| PatchTST + LayerNorm (MAE) | 0.209 | 0.301 | 0.412 | 0.713 |
| PatchTST + RMSNorm (MAE) | 0.209 | 0.300 | 0.411 | 0.715 |
| Last-value predictor (MSE) | 0.081 | 0.167 | 0.306 | 0.810 |
| PatchTST + UnitNorm (MSE) | **0.085** | **0.178** | **0.321** | **0.861** |
| PatchTST + BatchNorm (MSE) | 0.088 | 0.180 | 0.336 | 0.874 |
| PatchTST + LayerNorm (MSE) | 0.091 | 0.180 | 0.324 | 0.901 |
| PatchTST + RMSNorm (MSE) | 0.091 | 0.179 | 0.322 | 0.904 |

## F    Comparison with Last-Value Predictor

To provide a more comprehensive evaluation of UnitNorm's effectiveness, we compared it against a simple but effective baseline: the last-value predictor. This baseline simply uses the last observed value as the prediction for all future time steps, which can be surprisingly effective for time series with slow-changing patterns or high levels of noise.

Tables T1 and T2 present the comparison between the last-value predictor and PatchTST model with different normalization methods on the Exchange and ETTh2 datasets, respectively.

On the Exchange dataset (Table T1), the last-value predictor outperforms all Transformer-based models. This is likely due to the highly stochastic nature of exchange rates, where the most recent value is often the best predictor for future values. However, among the Transformer models, UnitNorm consistently achieves the best performance across all prediction horizons for both MSE and MAE metrics. This demonstrates that even when sophisticated models struggle to outperform simple baselines on challenging datasets, UnitNorm still provides relative advantages over other normalization techniques.

On the ETTh2 dataset (Table T2), all Transformer models substantially outperform the last-value predictor, with UnitNorm showing the best overall performance. UnitNorm achieves the best MSE scores across all prediction horizons and the best MAE for horizons 96, 192, and 336. This confirms UnitNorm's ability to enable Transformer models to effectively capture periodic patterns present in this dataset, as indicated by the higher periodicity scores shown in Table T4.

These results highlight an important insight: UnitNorm's effectiveness is most pronounced on datasets with strong periodic components (like ETTh2), while its advantages may be less significant on datasets where complex models are less suitable overall (like Exchange). This aligns with our theoretical analysis that UnitNorm helps preserve attention distributions and mitigate token shift issues, which are particularly beneficial for modeling periodic patterns in time series data.

Table T2: Last-value predictor versus PatchTST with different normalization methods on ETTh2 dataset. Best results are in **bold**.

| Model & Metric\Prediction | 96 | 192 | 336 | 720 |
|---|---|---|---|---|
| Last-value predictor (MAE) | 0.422 | 0.473 | 0.511 | 0.519 |
| PatchTST + UnitNorm (MAE) | **0.341** | **0.394** | **0.429** | 0.449 |
| PatchTST + BatchNorm (MAE) | 0.346 | 0.399 | 0.435 | 0.453 |
| PatchTST + LayerNorm (MAE) | 0.345 | 0.399 | 0.431 | **0.447** |
| PatchTST + RMSNorm (MAE) | 0.345 | 0.399 | 0.431 | 0.448 |
| Last-value predictor (MSE) | 0.432 | 0.534 | 0.597 | 0.594 |
| PatchTST + UnitNorm (MSE) | **0.290** | **0.369** | **0.417** | **0.429** |
| PatchTST + BatchNorm (MSE) | 0.292 | 0.376 | 0.421 | 0.435 |
| PatchTST + LayerNorm (MSE) | 0.297 | 0.380 | 0.417 | 0.429 |
| PatchTST + RMSNorm (MSE) | 0.297 | 0.379 | 0.418 | 0.429 |

## G    SELECTION OF HYPERPARAMETER $k$

The hyperparameter $k$ in UnitNorm plays a crucial role in determining the sparsity of attention scores and consequently affects model performance. To empirically identify the optimal $k$ values for time series applications, we conducted extensive experiments with various $k$ values on multiple datasets.

### G.1    EXPERIMENTAL SETUP

We performed a comprehensive sweep of fixed $k$ values ($k = 0.1, 0.3, \ldots, 0.9, 1.1$) on the ETTh2 datasets using the PatchTST model. We trained the model with different prediction horizons (96, 192, 336, and 720 time steps) and measured the Mean Square Error (MSE). All experiments were conducted with 3 different random seeds (41, 42, and 43), and the reported results are averaged across these runs.

### G.2    RESULTS ON ETTH2

Table T3 presents the MSE performance of PatchTST on the ETTh2 dataset with different fixed $k$ values. The results show that the optimal $k$ value depends on the prediction horizon, but generally falls within the range of 0.5 to 0.7. For the ETTh2 dataset, $k = 0.7$ achieves the best performance for prediction horizons of 96, 336, and 720, while $k = 0.5$ performs best for the horizon of 192.

### G.3    ANALYSIS

Our experiments reveal several important findings regarding the selection of $k$:

1. **Performance Robustness**: The performance of UnitNorm is generally not highly sensitive to small perturbations in the $k$ value. This is evident from the relatively small percentage differences between adjacent $k$ values, especially in the 0.5 to 0.7 range.

2. **Optimal Range**: The optimal $k$ value consistently falls within the range of 0.5 to 0.7 across different prediction horizons and datasets. This range provides a good balance between attention diversity and focus.

3. **Performance Gain**: The appropriate selection of $k$ can lead to moderate but meaningful performance improvements. For example, using $k = 0.7$ instead of

Table T3: MSE of PatchTST on ETTh2 with different fixed $k$ values. The relative difference from the best performing $k$ value is shown in parentheses. The best performing $k$ for each prediction length is highlighted in **bold**.

| $k$ \ prediction length | 96 | 192 | 336 | 720 |
|---|---|---|---|---|
| 0.1 | 0.2897 (+0.633%) | 0.3716 (+1.866%) | 0.4148 (+0.3%) | 0.4278 (+0.787%) |
| 0.3 | 0.293 (+1.769%) | 0.3668 (+0.535%) | 0.414 (+0.088%) | 0.4314 (+1.629%) |
| 0.5 | 0.29 (+0.719%) | **0.3648 (+0.0%)** | 0.4197 (+1.469%) | 0.4252 (+0.168%) |
| 0.7 | **0.2879 (+0.0%)** | 0.3717 (+1.896%) | **0.4136 (+0.0%)** | **0.4245 (+0.0%)** |
| 0.9 | 0.2967 (+3.069%) | 0.3764 (+3.187%) | 0.4175 (+0.937%) | 0.4283 (+0.904%) |
| 1.1 | 0.2987 (+3.744%) | 0.365 (+0.057%) | 0.4188 (+1.265%) | 0.435 (+2.472%) |

$k = 1.1$ for the 720-step prediction horizon on ETTh2 results in a 2.5% reduction in MSE.

4. **Dataset Dependence**: The optimal $k$ value is somewhat dataset-dependent, reflecting the varying degrees of periodicity and other temporal patterns across different time series data.

### G.4 Practical Recommendations

Based on our findings, we offer the following practical recommendations for selecting $k$ values:

- For datasets with strong periodicity, starting with $k = 0.7$ is recommended as it generally provides good performance across various prediction horizons.
- For datasets where the optimal $k$ is uncertain, we recommend trying both $k = 0.5$ and $k = 0.7$ to determine which works better for the specific application.
- In scenarios requiring maximum flexibility, implementing UnitNorm with a learnable $k$ parameter allows the model to adaptively determine the optimal attention sparsity during training.
- For comprehensive optimization, a validation-based approach can be employed where different fixed $k$ values are evaluated on a validation set to select the best configuration.

## H Extended Experimental Results

This section provides additional experimental results and analyses that supplement the main experiments presented in Section 4.

### H.1 Periodicity Measurement Analysis

Time series periodicity is a crucial factor in forecasting performance, especially for long-term predictions. To quantify the periodic patterns present in our experimental datasets, we conducted a comprehensive periodicity analysis using normalized correlation between input and expected output series.

As shown in Table T4, the datasets exhibit varying degrees of periodicity across different prediction horizons. ETTh2 consistently shows the strongest periodic patterns (with scores around 0.41-0.42), followed by Exchange (0.33-0.40) and ETTh1 (0.34-0.35), while ECL demonstrates the weakest periodicity (0.16-0.17).

Interestingly, the Exchange dataset shows an increasing trend in periodicity as the prediction horizon extends, reaching its highest correlation value (0.396) at the 720-step horizon. This suggests that longer-term patterns become more apparent in financial exchange data when viewed over extended time frames.

Table T4: Periodicity measurement on datasets. Scores are given as the maximum normalized correlation between input and expected output series. Higher scores indicate stronger periodic patterns that may be leveraged by forecasting models.

| Dataset\Prediction | 96 | 192 | 336 | 720 |
|---|---|---|---|---|
| ETTh1 | 0.347 | 0.351 | 0.353 | 0.338 |
| ETTh2 | 0.404 | 0.415 | 0.420 | 0.413 |
| ECL | 0.156 | 0.162 | 0.165 | 0.158 |
| Exchange | 0.328 | 0.340 | 0.353 | 0.396 |

The ability of UnitNorm to maintain consistent attention distributions, as discussed in Section 3, makes it particularly well-suited for capturing these periodic patterns, especially in datasets with stronger periodicity. This helps explain UnitNorm's superior performance on datasets like ETTh2 and Exchange in the long-term forecasting experiments.

### H.2 Performance on Large-Scale Dataset with Modern Architecture

To evaluate UnitNorm's effectiveness beyond standard benchmarks, we conducted additional experiments using the Pathformer model on the Solar dataset, which represents a significant increase in scale and complexity compared to our other test datasets.

**Solar Dataset** The Solar dataset is a large-scale time series collection containing data from 137 photovoltaic power plants across the United States, with approximately 52,000 samples. Its high dimensionality and real-world nature make it an excellent test case for evaluating normalization techniques in complex, practical scenarios.

**Pathformer Model** Pathformer is a state-of-the-art time series Transformer architecture that introduces innovations in handling multivariate time series through path-wise modeling. Unlike traditional Transformer architectures, Pathformer incorporates specialized mechanisms for capturing both temporal and cross-series dependencies.

Table T5: Pathformer performance on the Solar dataset (large-scale, 137 channels, 52K samples) with prediction horizon of 96. Results are averaged over 3 random seeds with standard deviation reported. Best results are in **bold**; second best are in *italic*. These results demonstrate UnitNorm's effectiveness even with modern architectures and large-scale datasets.

| Normalization | MSE | MAE |
|---|---|---|
| BatchNorm | $0.2215 \pm 0.0080$ | $0.2075 \pm 0.0121$ |
| LayerNorm | $0.2177 \pm 0.0064$ | $\mathbf{0.1996 \pm 0.0100}$ |
| RMSNorm | $0.2225 \pm 0.0031$ | $0.2097 \pm 0.0151$ |
| UnitNorm ($k = 0.0$) | $0.2202 \pm 0.0038$ | $0.2062 \pm 0.0082$ |
| UnitNorm ($k = 0.5$) | $\mathbf{0.2176 \pm 0.0041}$ | *$0.2053 \pm 0.0181$* |
| UnitNorm ($k = 0.7$) | *$0.2177 \pm 0.0091$* | $0.2074 \pm 0.0133$ |
| UnitNorm ($k = 1.0$) | $0.2267 \pm 0.0092$ | $0.2074 \pm 0.0054$ |

**Result Analysis** As shown in Table T5, UnitNorm with $k = 0.5$ achieves the best MSE ($0.2176 \pm 0.0041$) and competitive MAE ($0.2053 \pm 0.0181$, second only to LayerNorm). Several observations are worth noting:

- UnitNorm's performance remains strong even with more complex model architectures and larger datasets, demonstrating its generalizability.

- The optimal value of $k$ for this dataset appears to be around 0.5-0.7, which aligns with our findings in the main experiments.
- While LayerNorm achieves the best MAE, UnitNorm provides more balanced performance across both MSE and MAE metrics.
- Compared to RMSNorm and BatchNorm, UnitNorm consistently delivers superior results, reinforcing its advantages over these traditional normalization methods.

These results further substantiate UnitNorm's potential as a broadly applicable normalization technique for time series analysis tasks across various model architectures and dataset scales. The consistent performance on the Solar dataset, with its high dimensionality and large sample size, suggests that UnitNorm's benefits extend to real-world, large-scale applications.

