# OpenReview forum: "UnitNorm: Rethinking Normalization for Transformers in Time Series"
_ICLR.cc/2026/Conference — ICLR 2026 Conference Withdrawn Submission_

### Official Review · Reviewer_8fXg · 2025-10-30

**Soundness:** 3
**Presentation:** 2
**Contribution:** 2
**Rating:** 4
**Confidence:** 3

**Summary:**

This paper analyzes the drawbacks of widely used normalization approaches in Transformers, such as LayerNorm and BatchNorm, and proposes a new normalization method named UnitNorm for time series analysis (TSA) tasks. The paper provides theoretical analysis to explain how conventional normalization methods may cause issues such as token shift, attention shift, and sparse attention. Experimental results show that UnitNorm achieves superior performance on forecasting, classification, and anomaly detection tasks.

**Strengths:**

- The paper provides thorough and well-motivated theoretical analysis regarding the limitations of conventional normalization methods, identifying three potential issues—token shift, attention shift, and sparse attention—that could degrade Transformer performance on time series tasks.
- The effectiveness of the proposed UnitNorm is validated across diverse time series applications, including forecasting, classification, and anomaly detection. The experimental results demonstrate that the UnitNorm with proper parameter settings achieves superior performance compared to widely used normalization techniques.

**Weaknesses:**

- Some claims lack sufficient justification:
  - In *line 123*, the statement *“altering token vector orientations and obscuring long-range dependencies”* is vague. It would be helpful to provide clearer theoretical or empirical evidence explaining why standard normalization methods cause this issue.
  - In *line 267*, the paper claims that sparse attention is problematic in TSA tasks, but this claim is not sufficiently explained or analyzed. Additional clarification or supporting experiments would strengthen this argument.

- The motivation for focusing exclusively on time series analysis (TSA) is not entirely clear. The identified issues—token shift, attention shift, and sparse attention—may also arise in other Transformer-based applications (e.g., NLP or vision). It would be beneficial to discuss whether UnitNorm could generalize beyond TSA tasks.
- The proposed solution may lack of technical contribution. The proposed UnitNorm formulation appears to be a relatively minor modification of RMSNorm, differing mainly by a scaling factor. The paper would benefit from a clearer discussion of the fundamental difference and why such a seemingly small change leads to consistent improvements.
- The experiments indicate that UnitNorm’s performance is sensitive to its hyperparameter $k$, which could limit its practical applicability. Providing a heuristic guideline for selecting $k$ would make the method more practical.
- It is recommended to place the important experiments in the main text instead of the appendix.

**Questions:**

Please refer to the weaknesses part.

---

### Official Review · Reviewer_MaWK · 2025-10-31

**Soundness:** 3
**Presentation:** 3
**Contribution:** 2
**Rating:** 4
**Confidence:** 4

**Summary:**

This paper identifies three issues with center-and-scale normalizations (LayerNorm[1], BatchNorm[2]) in time series Transformers: (1) token shift-centering can flip dot-product signs, (2) attention shift-altering relative importance of tokens, and (3) sparse attention-reducing entropy of attention distributions. UnitNorm is proposed as a scale-only alternative. Theoretical analysis includes a sign-flip probability bound (Theorem 2.1) and an entropy lower bound (Theorem 3.2). Experiments across forecasting (4 datasets), classification (4 datasets), and anomaly detection (1 dataset) with 6 Transformer architectures show UnitNorm often outperforms LayerNorm/BatchNorm in average rank, with notable gains on periodic datasets (ETTh2: -1.46 MSE). However, comparisons miss key time-series normalization baselines (RevIN[3]), and improvements are inconsistent across datasets.

**Strengths:**

1. A minimal, easily implementable normalization layer, drop-in for many backbones. (The code is provided with reproducible information.)
2. Links token scale to attention entropy/sparsity with interpretable bounds. This helps reason about when cantering may hurt.
3. Linking normalization to attention entropy via Theorems 3.2-3.3 is insightful and not addressed in prior work. The entropy lower bound provides actionable guidance.
4. Covers multiple architectures and tasks, including forecasting, classification, and anomaly detection, with signs of benefit at longer horizons (6 architectures × 10 datasets × 3 tasks).
5. Synthetic and probing analyses that visualize how normalization affects token geometry and attention distributions.

**Weaknesses:**

1. Missing time-series baselines for non-stationarity. No comparison with RevIN (input-level reversible instance normalization) and related methods (e.g., DAIN[4]) that explicitly target distribution/regime shifts-central to the paper’s motivation.
2. Incremental novelty vs. RMSNorm[3]. Algorithmically close to RMSNorm (scale-only) with a fixed modulus; the core novelty is the explicit norm target and entropy framing. More distinctive empirical behavior is needed to clear a top-tier bar.
3. k-parameter and practical novelty. The proposed approach is presented as a tunable factor modulating attention entropy. However, the paper’s results indicate that fixed values (0.5 – 0.7) are empirically optimal, while learnable values yield no consistent gain. As a result, behaves as a hyper-parameter rather than a learnable control variable. This weakens the claim of architectural innovation, since without adaptive learning or systematic tuning analysis, UnitNorm functions as RMSNorm with an extra scalar rescaling. Demonstrating that a learnable consistently improves training stability or OOD robustness would strengthen the contribution.
4. Inconsistent effect sizes. Improvements are not universal across datasets/tasks; claims of generality would benefit from per-dataset significance tests and analyses correlating gains with data properties (seasonality, trend, shift severity).
5. Ablations are not fully developed. How sensitive is performance to? Is learning beneficial/stable? Interactions with Pre-LN vs post-LN, residual scaling, and long-sequence depth are not fully explored.

**Questions:**

1. Authors can consider adding RevIN (and, if possible, DAIN) across the same backbones/splits. Since these are designed for non-stationarity, they are the decisive baselines for your motivation.
2. Can you include regime-split evaluations (e.g., train on one market regime/weather season, test on another), rolling-origin time-shifted OOD, and cross-dataset transfer to verify robustness claims?
3. Can you show curves of performance vs. and attention entropy vs. Does learning consistently beat fixed? Any stability issues?
4. Have you tried UnitNorm at input embeddings (like RevIN), within MHA/FFN, or only at the first layer with re-projection at the end? Any insight on optimal placement(s) and interactions with residual scaling?
5. Can you report wall-clock/runtime and memory deltas, or anything that implies the efficiency or complexity to compare with baselines?

---

### Official Review · Reviewer_kJDv · 2025-10-31

**Soundness:** 2
**Presentation:** 3
**Contribution:** 1
**Rating:** 4
**Confidence:** 4

**Summary:**

The authors propose UnitNorm, a normalization method that scales input vectors by their norms without centering and introduces a scaling factor k. The authors argue that conventional normalization methods (LayerNorm, RMSNorm, etc.) may cause token shift, attention shift, and sparse attention problems in Transformers—especially for time-series forecasting—due to sign flips in normalized representations.
The authors provide theoretical analysis (probability of sign flips, gradient invariance, entropy lower bounds), and demonstrate empirical results across forecasting, classification, and anomaly detection tasks.
While the method is conceptually sound and well-motivated, its empirical validation is limited, especially given the emergence of large-scale time-series foundation models (Chronos2, Moirai, Sundial). As a result, the paper’s contribution does not yet meet the bar for such strong generalization claims.

**Strengths:**

1. The approach is well motivated, clearly identifies normalization–attention interactions as a key issue, framing token/attention shift as a quantifiable phenomenon.
2. Solid theoretical analysis — Includes formal theorems (sign-flip probability, gradient invariance, entropy bound) with proofs in the appendix, giving strong mathematical grounding.
3./Implementation simplicity — UnitNorm can directly replace existing normalization layers, making it easy to adopt.

**Weaknesses:**

1. Insufficient empirical validation on large-scale datasets and models. The paper mainly uses small to medium-sized benchmarks (ETTh1/2, ECL, Exchange, Solar). However, recent works such as Chronos2, Moirai, and Sundial have trained on large-scale datasets (LOSTA e.g.), together with the large-scale, multi-domain benchmarks (GIFT-EVAL, FEV) that test model robustness and scalability. Without evaluation on these, the claim that UnitNorm “generalizes across time-series domains” is not sufficiently supported.
2. Theoretical assumptions may not hold in real data.  In real world, time series are often non-Gaussian, autocorrelated, and heteroskedastic. The paper does not provide empirical evidence that sign flips actually occur frequently in practice or that they correlate with performance drops.

**Questions:**

see weaknesses

---

### Official Review · Reviewer_wABE · 2025-11-01

**Soundness:** 2
**Presentation:** 2
**Contribution:** 2
**Rating:** 2
**Confidence:** 4

**Summary:**

This paper revisits normalization in time-series Transformers and identifies theoretical issues with centering-based normalization methods, such as token-direction distortion and attention misalignment. It introduces UnitNorm, norm-based normalization method keeping vector directions and provides a higher entropy of attention values, enabling models capturing periodicity in time series. The paper combines theoretical analysis, synthetic dataset experiments, and results on multiple time-series tasks to argue that UnitNorm stabilizes attention geometry and yields gains over existing normalization layers.

**Strengths:**

- Normalization in time-series Transformers is an under-examined but increasingly important research direction.
- The discussion on vector orientation, sign-preservation, and attention fidelity provides useful intuition to the field.
- Evaluation spans multiple time-series tasks: forecasting, classification, and anomaly detection.

**Weaknesses:**

- PatchTST, FEDFormer, and CrossFormer performances in this paper are significantly weaker than official numbers, raising validity concerns for all reported gains.
- PatchTST relies critically on RevIN (Kim et el., ICLR 2022), yet it is unclear whether RevIN is applied fully or consistently. This is a major concern because the presence or absence of RevIN fundamentally alters the statistical properties of time-series input (scale distribution, domain-shift behavior, and vector directions), thereby substantially impacting PatchTST performance and training stability.
- The use of bold and underline in Table S9 appears inconsistent, making it unclear how to interpret the reported results. Please revise for clarity and consistency.
- In Table S7, UnitNorm is evaluated across multiple values of k. While it is reasonable that UnitNorm requires selection of k, presenting multiple k values during comparison complicates the fairness and interpretability of the results. This presentation also suggests that UnitNorm may be sensitive to the choice of k. Notably, even with multiple k configurations, LayerNorm or RMSNorm still outperform UnitNorm in several cases.
- If the authors intend to write classification and anomaly-detection results into the main scripts, then the corresponding results for those tasks should also be included in the main text for completeness and consistency.

**Questions:**

- In Figure 5, the y-axis is described as “average rank over models,” yet the values are shown in the range of 0 to 1. It is unclear how a rank metric can yield values in this range. Could the authors clarify how the rank is computed, and whether this reflects a normalized rank, or some others?
- The paper does not specify which models were included to compute the “average rank over models” in Figure 5. Please explicitly list the models used for this aggregation, as the lack of this information makes the figure difficult to interpret and reproduce.

---

### Note · Authors · 2025-12-04

I have read and agree with the venue's withdrawal policy on behalf of myself and my co-authors.